# QuBS: Combinatorial Quantum Brain Surgeon on Quantum Annealers

## Abstract

Quantum Neural Networks (QNNs) are parameterized quantum circuits trained for machine learning tasks. Sparse QNNs are generally desired since: 1) their limited entanglement topologies are less likely to approximate 2-design unitaries, thereby relieving the measurement concentration problem, which makes output statistics uninformative; and 2) they reduce susceptibility to decoherence and cumulative operational errors—a critical requirement for high-fidelity execution on near-term quantum hardware. This necessitates a method to prune QNNs with minimal fidelity loss. Existing techniques predominantly rely on Euclidean-space heuristics, such as Pauli rotation strengths or their gradients, to serve as proxies for QNN operation importance. However, these approaches are misaligned with the Hilbert-space dynamics and fundamentally ignore the non-commutativity of quantum operations (i.e. inter-parameter correlations). In response to such limitations, we propose *QuBS*, a novel QNN pruning framework that explicitly accounts for the QNN's Hilbert-space evolution. The core of *QuBS* involves searching for an optimal sparse subnetwork by solving a Quadratic Unconstrained Binary Optimization (QUBO) problem leveraging quantum annealers; this search is performed on the order of microseconds per anneal on the quantum hardware, enabling rapid sampling of the solution space. We further propose an iterative variant of *QuBS* to enable pruning QNNs with less annealer resource requirements. We evaluate *QuBS* on pre-trained QNNs designed for image classification and benchmark it against established protocols. We show that *QuBS* consistently achieves substantially lower fidelity loss during pruning; this translates to up to 30 percent improvement in retained classification accuracy for our image classification task at high sparsity levels compared to existing baseline methods.

## 1 Introduction

Quantum Neural Networks (QNNs) have emerged as potential candidates for resource-efficient machine learning, yet their practical utility remains constrained (Preskill, 2018; Zhang et al., 2023; Krinner et al., 2022) for applications such as physics simulations (Bauer et al., 2020; Gibbs et al., 2024) and visual computing (Wang et al., 2025) when the circuit structure is overly complex. Inspired by classical neural network pruning foundations such as Optimal Brain Damage (LeCun et al., 1989) and Optimal Brain Surgeon (Hassibi & Stork, 1992), QNN pruning aims to relief these challenges by removing redundant quantum operations while maintaining high operational fidelity. Existing QNN pruning methods typically rely on Euclidean metrics, such as evolution-strength magnitudes (Sajadimanesh et al., 2025) or gradient norms (Kulshrestha et al., 2024), to estimate gate importance. *However, these heuristics do not reflect how quantum states evolve in the Hilbert space and provide no general guarantee on preserved performance. Their underlying assumptions—parameter independence and gate commutativity—fail to capture the inherently non-linear, non-commutative nature of gate removal.* This leads to substantial performance degradation.

Motivated by these limitations, we introduce QuBS, a novel framework for combinatorial QNN pruning; see Fig. 1 for the scheme. QuBS formulates the pruning objective as a second-order approximation of the fidelity loss due to gate removal, thereby casting the challenge of identifying an optimal sparse circuit as a Quadratic Unconstrained Binary Optimisation (QUBO) problem. We show that the QUBO formulation can be efficiently constructed via parallelised forward-only QNN

Figure 1: *QuBS* formulates QNN pruning as a QUBO problem The logical problem is mapped onto the physical annealer topology through minor embedding. The lowest-energy sample returned by the annealer encodes a sparse QNN topology with minimal fidelity drop $F(|\psi_d\rangle, |\psi_s\rangle) = |\langle\psi_d|\psi_s\rangle|^2$.

evaluations. Then, the NP-hard complexity of QUBOs (challenging for classical solvers) motivates our use of contemporary noisy quantum annealers—an emerging class of Ising machines that natively solve QUBO problems through adiabatic quantum evolution and hold promise for discovering high-quality solutions more efficiently for problems characterised by rugged energy landscapes (Denchev et al., 2016; Kadowaki & Nishimori, 1998). Since monolithic, global pruning can exceed current annealer resource capacities, we further propose an iterative variant of QuBS that enables progressive, local pruning, which collectively approximates a global optimum with lower hardware requirements. To summarise, the technical contributions of this work include:

- *QuBS*, the first end-to-end framework that leverages available quantum annealers for combinatorial QNN pruning with theoretical analysis (Sec. 5);

- A principled and novel QUBO formulation that integrates a Lagrangian-form soft penalty to enforce a target sparsity. The lowest energy sample encodes the optimal sparse sub-QNN topology (Sec. 5.3);

- An iterative variant of *QuBS* that performs tractable, progressive QNN pruning, enabling QNN pruning at lower hardware requirements (Sec. 5.5).

We empirically evaluate the performance of *QuBS* on pre-trained QNNs designed for image classification; see Fig. 2 for the scheme. **In this context, the preserved fidelity of the sparse QNN directly governs its retained classification accuracy.** *Note that we focus on QNN pruning assuming ideal hardware and do not consider noise effects; it is a direction orthogonal to the contributions of this work.* Our results demonstrate that *QuBS* consistently outperforms the best-performing baseline, preserving up to approx. 30 percent more classification accuracy after pruning, a substantial margin that underscores the efficacy of our approach in identifying high-fidelity sparse circuit topologies; see Sec. 6.

## 2 RELATED WORK

**QNN Pruning**. Quantum circuits characterized by dense topologies are susceptible to measurement concentration (McClean et al., 2018; Cerezo et al., 2020), and face deployment challenges on NISQ devices (Preskill, 2018; Yan et al., 2023). In response, a growing body of research investigates into making the QNN sparse. Hu et al. (2022) introduce QNN pruning and analyze its behavior on preserving execution fidelity. Sajadimanesh et al. (2025) proposes NR-QNN, a method that gauges the quantum evolution importance with its strength, while Kulshrestha et al. (2024) argues that the quantum evolution gradient is a better proxy and introduces QAdaPrune based on the principle; to the best of our knowledge, these are the only existing methods that address the general QNN pruning problem that we target. We summarise

| Property | *QuBS* (Ours) | NR-QNN | QAdaPrune |
|---|---|---|---|
| Pruning Theory | Approx. Fidelity Drop | Evolution Strength | Evolution Gradient |
| Pruning Target | Circuit Gate/ Depth | Circuit Gate | Circuit Gate |
| Commutation-aware | ✓ (2nd order) | ✗ | ✗ |
| QNN Platform | Simulator (w/o noise) | Simulator (w/o noise) | Simulator (w/o noise) |

Table 1: Comparative algorithmic analysis for QNN pruning. "Approx." means approximated.

Figure 2: A representative scheme of our learnable QNN for image classification. The parametrized quantum circuit evolves the encoded image features, followed by projective measurements into the computational basis. The resulting binary measurements are mapped to discrete class labels.

key algorithmic differences with ours in Tab. 1. *An orthogonal direction: quantum architecture search, lies in searching for optimal task-specific QNN architecture, while QNN pruning simplifies pre-existing QNN topologies by removing quantum operations with minimum fidelity drop.*

**Annealing-based Quantum Optimizations**. Leveraging quantum annealers as efficient solvers for combinatorial optimization problems was first introduced by Farhi et al. (2001). Since then, quantum annealers have been applied to various problems such as point set alignment, shape matching, and multi-object tracking (Golyanik & Theobalt, 2020; Birdal et al., 2021; Zaech et al., 2022; Farina et al., 2023; Benkner et al., 2021; Arrigoni et al., 2022), primarily through manually encoding and transforming these problems into Ising coupling forms executable on annealers. Recently, Benkner et al. (2023) proposed a data-driven approach, QuAnt, that learns the QUBO formulations directly from data within specific problem scenarios, enabling flexible and compact solution encodings.

Our work draws inspiration from Yulianti et al. (2024). While they target selecting an optimal ensemble of neural expert models by formulating it as a binary optimization problem, we focus on the distinct challenge of combinatorial QNN pruning to identify a sparser sub-QNN while preserving operational fidelity. *We share the computational strategy of leveraging quantum annealers as efficient QUBO solvers.*

## 3 REVIEW: QNN AND ADIABATIC QUANTUM COMPUTATION (AQC)

This section reviews the fundamental concepts of QNN and AQC.

**Parametrized Quantum Gates as Neural Connections.** A QNN implements a unitary transformation $U(\mathbf{w}) = \prod_{i=1}^{N} e^{-iw_i G_i}$ where each $w_i \in \mathbb{R}$ is a tunable parameter and $G_i$ is a Hermitian generator. Single-qubit rotations generated by Pauli operators take the general form $R_\alpha(w) = e^{-iw\sigma_\alpha/2}$, $\alpha \in \{X, Y, Z\}$, and parametrized entangling gates can be written as $CR_\alpha(w) = |0\rangle\langle 0| \otimes I + |1\rangle\langle 1| \otimes R_\alpha(w)$ These gates act as quantum analogues of neural connections, with the circuit architecture determined by the ordered set of generators $\{G_i\}$. Gate removal corresponds to parameter nullification, since $\lim_{w\to 0} e^{-iwG} = I$ ($I$ is the identity matrix), allowing pruning to be expressed as selectively setting parameters to zero while maintaining the unitary structure of the circuit.

**Adiabatic Quantum Computation and QUBO.** Adiabatic quantum computation solves optimization problems by evolving a system from an initial Hamiltonian $H_0$ to a problem Hamiltonian $H_P$ whose ground state encodes the desired solution. QUBO objectives of the form

$$\min_{x\in\{0,1\}^N} x^\top Q x + c^\top x \tag{1}$$

map naturally to Ising Hamiltonians via the standard correspondence $x_i = \frac{1}{2}(I - Z_i)$, where $Z_i$ is the Pauli-$Z$ operator. Substituting this identity yields

$$H_P = \sum_i h_i Z_i + \sum_{i<j} J_{ij} Z_i Z_j, \tag{2}$$

with $h_i$ and $J_{ij}$ determined by the QUBO matrix $Q$ and vector $c$. This is the native problem form supported by quantum annealers. After expressing QNN pruning as a QUBO problem, the annealer returns the lowest-energy configuration corresponding to the optimal sparse circuit.

## 4 SECOND-ORDER CORRELATIONAL STRUCTURES IN QNNS: THE ROLE OF COMMUTATIVITY

We begin with a preliminary analysis of the QNN pruning problem and identify conditions under which its complexity grows only linearly rather than exponentially. This analysis serves as the theoretical foundation supporting the usefulness of our proposed method. Consider a general QNN defined by a parametrized unitary transformation $U(\mathbf{w})$. The cost function $J(\mathbf{w})$ is typically taken as the expectation value of a Hermitian observable $H$ with respect to the circuit-prepared state $|\psi(\mathbf{w})\rangle$:

$$J(\mathbf{w}) = \langle \psi_0 | U^\dagger(\mathbf{w}) \, H \, U(\mathbf{w}) | \psi_0 \rangle, \tag{3}$$

where $|\psi_0\rangle$ is the input (reference) state and $|\psi(\mathbf{w})\rangle := U(\mathbf{w})|\psi_0\rangle$. Since many circuit parametrizations implement $U(\mathbf{w})$ as a product of parametrized gates $e^{-iw_iG_i}$, we often write

$$U(\mathbf{w}) = \prod_i e^{-iw_iG_i}, \tag{4}$$

with $G_i$ the Hermitian generator associated with parameter $w_i$. We show the second-order (correlational) structure of the composing parametrized QNN gates admits a compact expression:

**Proposition 1.** *Let $J(\mathbf{w})$ be defined as above and denote the Hessian by $\mathbf{H}$ with entries*

$$H_{ij}(\mathbf{w}) \;=\; \frac{\partial^2 J(\mathbf{w})}{\partial w_i \, \partial w_j}. \tag{5}$$

*Define the effective Hermitian generators*

$$\mathcal{G}_i(\mathbf{w}) := i \, U^\dagger(\mathbf{w}) \, \partial_{w_i} U(\mathbf{w}), \qquad \widetilde{\mathcal{G}}_i(\mathbf{w}) := U(\mathbf{w}) \, \mathcal{G}_i(\mathbf{w}) \, U^\dagger(\mathbf{w}). \tag{6}$$

*Then, the Hessian admits the exact representation*

$$H_{ij}(\mathbf{w}) = -\big\langle \psi(\mathbf{w}) \big| \big[\widetilde{\mathcal{G}}_j(\mathbf{w}), \, [\widetilde{\mathcal{G}}_i(\mathbf{w}), \, H]\big] \big| \psi(\mathbf{w}) \big\rangle \;+\; i \, \big\langle \psi(\mathbf{w}) \big| \big[\partial_{w_j}\widetilde{\mathcal{G}}_i(\mathbf{w}), \, H\big] \big| \psi(\mathbf{w}) \big\rangle. \tag{7}$$

*Proof.* See Appendix C for a more detailed derivation. $\square$

If the parametrization satisfies $\partial_{w_j}\widetilde{\mathcal{G}}_i(\mathbf{w}) = 0$, the Hessian simplifies to

$$H_{ij}(\mathbf{w}) = -\big\langle \psi(\mathbf{w}) \big| \big[\widetilde{\mathcal{G}}_j, \, [\widetilde{\mathcal{G}}_i, H]\big] \big| \psi(\mathbf{w}) \big\rangle.$$

In this reduced form, the off-diagonal elements $H_{ij}$ $(i \neq j)$ vanish precisely when the cross double commutators $[\widetilde{\mathcal{G}}_j, \, [\widetilde{\mathcal{G}}_i, H]]$ vanish. Thus it can occur in simple settings such as generators acting on disjoint subsystems with a correspondingly local observable—that all off-diagonal Hessian entries vanish. The pruning problem then reduces to an uncorrelated selection task with only linearly growing complexity instead of exponential. Such structural decoupling, however, is atypical for expressive QNNs, where nontrivial correlations between parameters generally persist, motivating our *QuBS* pruning framework for the general case, which we describe in the next section.

## 5 COMBINATORIAL QNN TOPOLOGY OPTIMIZATION

We next introduce *QuBS*, our combinatorial QNN pruning approach formulated as a QUBO problem. Our framework is computationally isomorphic to the Ising spin glass model, which enables native execution on quantum annealers. Sec. 5.1 first introduces the concept of individual gate pruning. Building on this, Sec. 5.2 extends the approach to the pruning of gate groups. Furthermore, Sec. 5.3 presents a method for enforcing sparsity constraints on the target QNN during pruning, while Sec. 5.4 analyzes the resulting implications due to such constraints for the QUBO formulation. Finally, Sec. 5.5 introduces an iterative, local pruning strategy as a scalable alternative to global monolithic pruning.

## 5.1 COMBINATORIAL QNN GATE-LEVEL PRUNING

Gate-level QNN topology pruning aims to construct a sparse circuit by treating quantum gates as the fundamental units for removal while minimizing operational fidelity drop. This process can be framed as a combinatorial optimization problem, searching over potential gate subsets to remove. We consider the combinatorial nature up to 2nd order, and the approximated fidelity change. The objective function $\mathcal{L}$ can then be expressed in a precise quadratic form, reducing QNN pruning to a QUBO problem. The lowest-energy sample obtained from the annealer is a binary string that minimizes the QUBO objective function; each binary value in this string explicitly encodes the retention or removal of a specific quantum gate, thereby defining the optimal sparse circuit topology. We outline the formulation details below:

**Proposition 2.** *The gate-level QNN pruning objective—minimizing the estimated fidelity loss from quantum gate removal—can be re-formulated as a QUBO problem in matrix form:*

$$\underset{\mathbf{m} \in \mathbb{B}^N}{\arg\min} \left[ \nabla \mathcal{L}(\mathbf{w}_0)^\top \delta \mathbf{w}_{pr} + \frac{1}{2} (\delta \mathbf{w}_{pr})^\top \nabla^2 \mathcal{L}(\mathbf{w}_0)(\delta \mathbf{w}_{pr}) \right] = \underset{\mathbf{m} \in \mathbb{B}^N}{\arg\min} \left( \mathbf{c}^\top \mathbf{m} + \mathbf{m}^\top \mathbf{Q} \mathbf{m} \right). \quad (8)$$

$\mathbf{m} \in \{0,1\}^N$ *is a binary mask defining the optimized quantum circuit topology with removed quantum gate ($m_i = 1$) or retained ones ($m_i = 0$). $\mathbf{w}_0$ denotes the quantum gate parameter and $\delta \mathbf{w}_{pr}$ describes the pruning-induced parameter change. The linear coefficient vector $\mathbf{c} \in \mathbb{R}^N$ and the symmetric quadratic coupling matrix $\mathbf{Q} \in \mathbb{R}^{N \times N}$ represent the first- and second-order combinatorial correlations among composing quantum operations:*

$$\mathbf{c} = - \begin{bmatrix} \cdots & w_0^i \frac{\partial \mathcal{L}}{\partial w^i}(\mathbf{w}_0) & \cdots \end{bmatrix}^\top, \quad \mathbf{Q} = \frac{1}{2} \begin{bmatrix} (w_0^1)^2 \frac{\partial^2 \mathcal{L}}{\partial (w^1)^2}(\mathbf{w}_0) & \cdots & w_0^1 w_0^N \frac{\partial^2 \mathcal{L}}{\partial w^1 \partial w^N}(\mathbf{w}_0) \\ \vdots & \ddots & \vdots \\ w_0^N w_0^1 \frac{\partial^2 \mathcal{L}}{\partial w^N \partial w^1}(\mathbf{w}_0) & \cdots & (w_0^N)^2 \frac{\partial^2 \mathcal{L}}{\partial (w^N)^2}(\mathbf{w}_0) \end{bmatrix} \quad (9)$$

*Proof.* See Appendix D. $\qquad \square$

The optimal mask $\mathbf{m}$ with the lowest energy obtained by sampling the equivalent Ising model on the annealer encodes the sparse QNN topology. The resulting circuit can then be compiled for execution.

## 5.2 COMBINATORIAL QNN DEPTH-LEVEL PRUNING VIA DEPENDENCY-AWARE GROUPING

By leveraging the inherent concurrency of unitary quantum operations acting on disjoint Hilbert subspaces and the non-Abelian nature of their interactions, we investigate reducing the effective circuit depth—formally defined as the minimal number of sequential time steps required for QNN execution under specific hardware connectivity constraints. This is equivalent to reducing the chromatic number $\chi(\Gamma)$ of the circuit's interaction graph $\Gamma = (V, E)$, where vertices represent gates $G_1, \ldots, G_N$ and edges connect gates that either (1) act on shared qubits or (2) are non-commuting ($[G_i, G_j] \neq 0$). We partition the gate set $G_1, \ldots, G_N$ into $M$ dependency-aware subsets $\mathcal{S}_1, \ldots, \mathcal{S}_M$; the partitioning is designed such that gates within each subset $\mathcal{S}_j$ can be executed concurrently (i.e. they share the same effective depth). A single binary decision variable $m_{\mathcal{S}_j} \in \{0, 1\}$ determines whether all quantum operations in $S_j$ are retained or removed to decrease circuit depth:

$$m_i = m_{\mathcal{S}_j}, \quad \forall G_i \in \mathcal{S}_j. \quad (10)$$

This group-wise enforcement maintains consistency of the dependency graph structure during pruning. The original gate-level QUBO problem (see Prop. 2) is reformulated in this reduced combinatorial space of depth groups. We present intermediate steps below.

**Proposition 3.** *The depth-level QNN pruning objective can be re-formulated as a QUBO problem defined over the reduced combinatorial space $\mathbf{m}' \in \{0,1\}^M$ of depth groups with updated linear $\mathbf{c}' \in \mathbb{R}^M$ and quadratic coefficients $\mathbf{Q}' \in \mathbb{R}^{M \times M}$ defined as:*

$$c_j' = \sum_{i \in I_j} c_i, \qquad Q_{jk}' = \sum_{i \in I_j} \sum_{l \in I_k} Q_{il}, \quad \forall j, k \in \{1, \ldots, M\}. \quad (11)$$

$\mathbf{m}' \in \{0,1\}^M$ *is a binary mask defining the optimized quantum circuit topology. A value $m_j' = 1$ indicates removal of the entire group (and thus reduction in the circuit effective depth), while $m_j' = 0$ indicates its retention. $I_j$ is the index set of gates in group $\mathcal{S}_j$. Coefficients $\mathbf{c}'$ and $\mathbf{Q}'$ represent combinatorial correlations among the QNN's effective depths.*

*Proof.* See Appendix E. □

## 5.3 SPARSITY-INFORMED QNN PRUNING VIA LAGRANGIAN-FORM SOFT PENALTY

The QUBO formulation derived thus far lacks an explicit mechanism to regulate the sparsity of the target QNN. To address this problem, we formalize the sparsity requirement by imposing a cardinality constraint on the binary mask $\mathbf{m}$ via a linear equality:

$$\mathbf{A}^\top \mathbf{m} = |\mathbf{m}|_0 = \mathbf{1}_N^\top \mathbf{m} = N_p = b, \tag{12}$$

where $\mathbf{1}_N \in \mathbb{R}^N$ is an all-one vector, and $N_p$ specifies the target number of QNN structures to be removed. We leverage Lagrangian duality to introduce this constraint into the original QUBO; see Prop. 1, by introducing a quadratic penalty term $\lambda(\mathbf{A}^\top \mathbf{m} - b)^2$. We present intermediate steps below:

**Proposition 4.** *The sparsity-informed gate-level QNN pruning objective:*

$$\underset{\mathbf{m} \in \mathbb{B}^N}{\arg\min} \left( \mathbf{c}^\top \mathbf{m} + \mathbf{m}^\top \mathbf{Q} \mathbf{m} \right) \quad s.t. \quad \mathbf{A}^\top \mathbf{m} = b \tag{13}$$

*can be re-formulated as an equivalent QUBO problem via Lagrangian-form soft penalty with the modified linear and quadratic coefficients:*

$$\tilde{\mathbf{c}} \in \mathbb{R}^N = (\mathbf{c}^\top - 2b\lambda \mathbf{A}^\top)^\top, \ \tilde{\mathbf{Q}} \in \mathbb{R}^{N \times N} = (\mathbf{Q} + \lambda \mathbf{A} \mathbf{A}^\top).$$

$\lambda > 0$ *is a penalty factor controlling the strictness of the constraint.*

*Proof.* See Appendix F. □

**Remark 1.** *The formulation naturally generalizes to depth-level QNN pruning by replacing QUBO coefficients in Eqn. 13 with those in Eqn. 11.*

While $\lambda$ must be sufficiently large to ensure asymptotic constraint satisfaction, excessively large values can distort the original QUBO objective landscape and impair solution quality; see detailed analysis in Zaborniak & Stege (2023). We will explain our choice of $\lambda$ in the experimental section.

## 5.4 CONVEXITY AND PENALTY-INDUCED SPECTRAL PERTURBATIONS OF $\tilde{\mathbf{Q}}$

While QUBO problems are NP-hard in general, their computational complexity can be reduced to polynomial when the coupling matrix $\tilde{\mathbf{Q}}$ is positive semi-definite (PSD) (Kochenberger et al., 2014). The induced convexity due to PSD collapses all local minima into a single global minimum and eliminates energy barriers—thereby negating the potential advantages tunneling effects provided by quantum annealing. We, therefore, analyze the spectral properties of the perturbed coupling matrix $\tilde{\mathbf{Q}} = \mathbf{Q} + \lambda \mathbf{A} \mathbf{A}^\top$. While $\mathbf{Q}$ may achieve asymptotic PSD near a locally optimal configuration upon sufficient training, *this property does not hold for arbitrary QNN in general; see dynamic QNN pruning (Sajadimanesh et al., 2025; Kulshrestha et al., 2024)*. The penalty term $\lambda \mathbf{A} \mathbf{A}^\top = \lambda \mathbf{1}_{N \times N}$ introduces a rank-1 spectral perturbation to $\mathbf{Q}$, inducing a structured perturbation of the eigenspectrum of the original coupling. This perturbation scales linearly with the penalty coefficient $\lambda$ and alters the optimization landscape through eigenvalue shifts while preserving the original eigenvectors. We formalize this spectral modification as follows:

**Proposition 5.** *Let $\mathbf{q} \in \mathbb{R}^N$ be an eigenvector of $\mathbf{Q} \in \mathbb{R}^{N \times N}$ with eigenvalue $\mu$. The perturbed eigenvalue $\mu'$ induced by the penalty term $\lambda \mathbf{A} \mathbf{A}^\top$ is characterized by*

$$\mu' = \mu + \lambda N \frac{|\langle \mathbf{A}, \mathbf{q} \rangle|^2}{\|\mathbf{A}\|^2 \|\mathbf{q}\|^2}. \tag{14}$$

*The perturbation strength is proportional to the alignment between $\mathbf{A}$ and $\mathbf{q}$.*

*Proof.* See Appendix G. □

## 5.5 ITERATIVE LOCAL COMBINATORIAL QNN PRUNING

We have introduced combinatorial QNN pruning from a global perspective, which considers all parametrized Hamiltonian evolutions simultaneously. For QNNs with complex circuit structures, the resulting QUBO problem dimension can exceed the hardware capacity of current-generation quantum annealers, making direct global QNN pruning intractable. We propose an iterative local QNN pruning strategy that decomposes the original high-dimensional QUBO into tractable subproblems; it is partially inspired by the sliding window protocol (Rowley et al., 1998). The approach leverages the insight that *any partial variable assignment $\{m_i\}_{i=1}^n \subset \{0,1\}^n$ (where $n < N$) induces a valid reduced-dimension QUBO instance for the remaining variables.* The QUBO coefficients $\mathbf{c}'' \in \mathbb{R}^{N-n}$ and $\mathbf{Q}'' \in \mathbb{R}^{(N-n)\times(N-n)}$ for the subproblem need to account for the iterative update:

**Proposition 6.** *Given original QUBO coefficients $\mathbf{c} \in \mathbb{R}^N$ and $\mathbf{Q} \in \mathbb{R}^{N \times N}$ from Eqn. 8, and a partial assignment $\mathbf{m} \in \mathbb{R}^n$, the reduced QUBO subproblem for the remaining $N - n$ decisions has the following coefficients instead:*

$$\mathbf{c}'' = \mathbf{c}[n:N] + 2\sum_{i=1}^n m_i \mathbf{Q}_{i,\cdot}, \quad \mathbf{Q}'' = \mathbf{Q}[n:N, n:N] \tag{15}$$

*where $\mathbf{Q}_{i,\cdot}$ contains the $i$-th row's last $N - n$ elements; $n$ is the window size.*

*Proof.* See Appendix H. □

The iterative local QNN pruning works as follows: 1) *priority metric construction and window sequencing*: we take a weak but globally-informed QNN pruning metric, allowing local structures (i.e. sliding window) to be processed with a priority order; in each iteration, we find 2) *local QUBO solution*: we perform QNN pruning for the current sliding window; see Eqn. 13, and fix the structures that have been pruned; 3) *dynamic QUBO reformulation*: we reformulate the reduced QUBO problem for the remaining structures according to Eqn. 15. The algorithmic details are described in Alg. 2.

**Remark 2.** *A large window size $n$ should be chosen within the available annealer's resource constraint, such that the returned QNN topology approximates better the global optimum.*

## 6 EXPERIMENTS

We evaluate our *QuBS* protocol on pruning QNNs during their training, analyzing the performance-sparsity trade-off with progressive circuit pruning. The QUBO problem—formulated with coupling terms and linear biases—is prepared on an Intel Xeon CPU (2.20 GHz, 12 GB RAM) and uploaded to D-Wave Advantage annealers; the process of logical-to-physical qubit embeddings and resulting resource allocations are visualized in Fig. 9 and Fig. 10. The annealing process is maintained at the default cryogenic temperature of 15 mK with a fixed 20 μs annealing time (excluding I/O latency). Each optimization cycle performs 100 reads to ensure statistical robustness, maintained uniformly across all trials. The Lagrangian penalty multiplier is set to $\lambda = 0.3$, determined via preliminary testing at maximum target sparsity to balance constraint satisfaction and objective fidelity; this value remains fixed through the experiments.

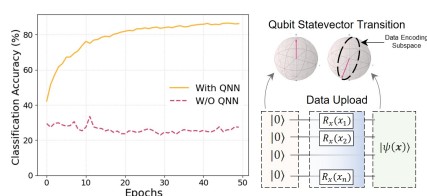

Figure 3: (Left) model's training process: classification performance on learnt features is visualized; (right) data are uploaded into a QNN via Pauli-X rotation.

**QNN Architecture and Training Protocol.** We implement a learnable quantum circuit inspired by Mitarai et al. (2018) to classify CIFAR-10 images (illustrated in Fig. 2). A convolutional feature extractor $\mathbf{z}_\phi(\mathbf{x})$ maps an input image $\mathbf{x}$ into a compact representation that is encoded into the quantum circuit; the parametrized quantum evolution $U(\boldsymbol{\theta})$ then transforms the encoded state, and measurement probabilities $P(\mathbf{y} \mid \mathbf{x}) = \left|\langle \mathbf{y} \mid U(\boldsymbol{\theta})\, \mathbf{z}_\phi(\mathbf{x})\rangle\right|^2$ are compared to one-hot labels $\mathbf{y}$ via a cross-entropy loss. The feature extractor and QNN are trained end-to-end on a high-performance Pennylane simulator (Bergholm et al., 2018) using hybrid optimization: classical parameters $\phi$ are updated by gradient descent while quantum parameters $\boldsymbol{\theta}$ are optimized with the parameter-shift rule; we use learning rate $\beta = 0.01$ and apply Gaussian initialization to the QNN (Zhang et al., 2022).

The complete data-encoding and training process is shown in Fig. 3. To evaluate *QuBS*, we perform combinatorial QNN pruning at two points in the 50-epoch schedule—early (15 epochs) and late (35 epochs)—so as to probe distinct training landscapes and obtain comprehensive insights into pruning behaviour. The QNN comprises $M = 5$ qubits and $P = 2$ strongly-entangling layers following the protocol of Schuld et al. (2020); each layer applies parameterized single-qubit rotations on every qubit followed by parameterized entangling operations between neighbouring qubits. A complete visualization of the circuit is provided in Fig. 4.

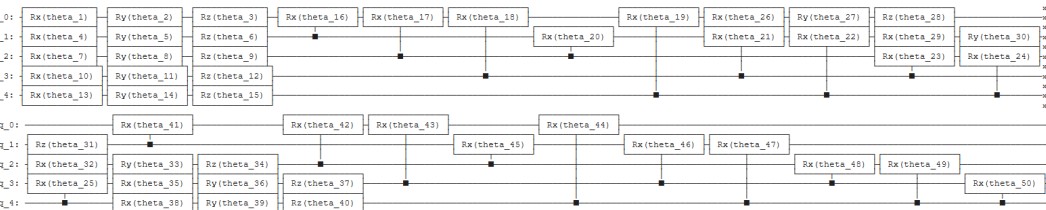

Figure 4: Visualization of the parametrized QNN in our experiments.

**Remark 3.** *The evaluated QNN does not claim architectural optimality, as (1) no standard QNN architecture exists for real-world tasks yet, and (2) QuBS focuses on combinatorial QNN pruning over its architectural design. It only serves as a testbed for our QNN pruning framework while QuBS is inherently agnostic to the chosen QNN architecture.*

## 6.1 COMBINATORIAL QNN GATE-LEVEL PRUNING

We evaluate *QuBS* to perform global combinatorial gate-level pruning, beginning with a carefully structured empirical study to investigate: 1) how well the second-order combinatorial QNN pruning can approximate the ground-truth solution which is computationally prohibitive to calculate for all but the smallest instances via exhaustive search; 2) how well can quantum annealers perform against its classical algorithmic benchmark simulated annealing; see Appendix J for details. We then evaluate *QuBS* on the QNN with 50 quantum gates for image classification following Alg. 1. We prepare the QUBO problem and solve it using D-Wave quantum annealers to obtain the pruning mask. The constraint-free and constraint-incorporated coupling matrices are visualized in Fig. 5, reflecting the structural impact of the Lagrangian penalty term. We benchmark *QuBS* against several established QNN pruning methods, including QAdaPrune (Kulshrestha et al., 2024) and NR-QNN (Sajadimanesh et al., 2025). Experimental results are visualized in Fig. 5. We observed that *QuBS* consistently outperforms existing methods across sparsity levels; the performance advantage becomes particularly significant under aggressive sparsity constraints. For example, at a sparsity ratio of $0.1$, we observed an improvement of the classification accuracy of $\approx 30$ and $50$ percent over the second-best approach. This resilience in high-sparsity regimes establishes *QuBS* as a compelling strategy for QNN pruning while maintaining operational fidelity. Some hardware analysis is provided, such as problem embeddings; see Appendix K, and the required hardware resources across problem scales; see Appendix L.

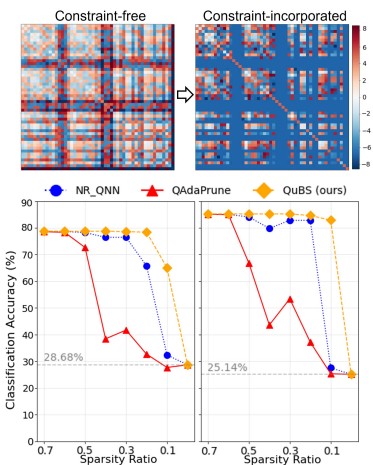

Figure 5: (Top) visualized penalty on the coupling; (bottom) *QuBS* evaluated on QNN models trained for: (left) 15, (right) 35 epochs against established methods.

## 6.2 COMBINATORIAL QNN DEPTH-LEVEL PRUNING

We perform global combinatorial pruning to minimize the effective circuit depth—defined as the minimal sequence of gate operations required for QNN execution; a scheme of effective circuit depth is provided in Fig. 6. The corresponding QUBO formulation for depth-level QNN pruning can be

constructed following Prop. 3 and 4; the problem is then up-
loaded to D-Wave machines to extract the optimal binary mask
that determines which depth groups to retain or remove. We
maintain identical experimental settings from gate-level QNN
pruning; experimental results are systematically visualized in
Fig. 6. The target QNN topology has an initial effective depth of
20 layers evaluated as described in Sec. 5.2; it serves as the base-
line for our depth reduction experiments. Across various target
depth reduction levels, we observed that *QuBS* outperforms
other methods, i.e. $\approx$30 percent improvement in maximum,
achieving robust circuit depth compression while preserving
operational fidelity.

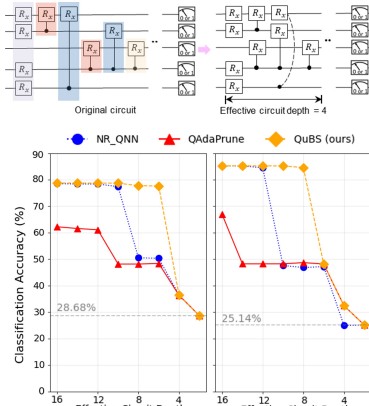

### 6.3 Complementary Study with Gurobi

We next experiment with Gurobi Gurobi Optimization, LLC
(2025), a leading commercial classical optimization solver. Us-
ing the unstructured QNN pruning setup from Fig. 5, we report
Gurobi's results in Table 2. A comparison with an advanced,
globally-oriented solver like Gurobi helps clarify the current
trade-off between solution quality and runtime. Because the
QUBO search space grows exponentially with the number of
binary variables, Gurobi's convergence is substantially slower

Figure 6: (Top): schematically de-
rived effective circuit; (bottom):
depth-level evaluation of *QuBS* on
QNNs trained for (left) 15 and
(right) 35 epochs benchmarked
against established methods.

than our QA solver, so we impose a 15-minute time limit per run; this pragmatic convergence
constraint is consistent with prior observations (Zaech et al., 2022). Empirically—while Gurobi
often finds solutions of slightly higher quality—the improvement is modest and comes at a markedly
greater computational cost. This complementary comparison also supports our forward-looking
motivation of using QA in QNN pruning, as the latter is expected to benefit from QUBO formulations
in future even more as quantum annealers continue to improve.

| Method | Epoch | Sparsity | | | | | | | | |
|--------|-------|------|------|------|------|------|------|------|------|------|
| | | 1.0 | 0.7 | 0.6 | 0.5 | 0.4 | 0.3 | 0.2 | 0.1 | 0.0 |
| QuBS | 15 | 78.7 | 78.7 | 78.7 | 78.7 | 78.7 | 78.5 | 78.4 | 65.0 | 28.6 |
| | 35 | 85.2 | 85.2 | 85.2 | 85.2 | 85.2 | 85.2 | 84.6 | 82.9 | 25.1 |
| Gurobi | 15 | 79.3 | 79.4 | 79.6 | 79.6 | 79.5 | 79.1 | 79.2 | 67.1 | 29.7 |
| | 35 | 86.1 | 86.2 | 85.3 | 86.4 | 86.1 | 86.1 | 85.3 | 83.3 | 27.0 |

Table 2: Comparative performance of QuBS using 1) a quantum annealer, and 2) the Gurobi solver,
across various sparsity levels.

### 6.4 Iterative Combinatorial QNN Gate- and Depth-level Pruning

Instead of globally pruning QNN topologies, we perform local iterative pruning with sliding windows
following Alg. 2. Motivated by the asymptotic convergence of quantum gates $U_k(w_k)$ to identity
operations $I$ on their target qubits under diminishing parameter magnitudes $\theta_k$ (Sajadimanesh et al.,
2025), i.e. $\lim_{\|w_k\| \to 0} U_k(w_k) = I$, we formalized a pruning priority metric: $\mathcal{P}(U_k(w_k)) = \|w_k\|_2$,
where a sliding window is defined, moving in the direction of ascending order of $\mathcal{P}$ to minimize
circuit distortion. Each iteration formulates the QUBO problem considering only the window-
bounded quantum operations while accounting for the second-order combinatorial effects of those
operations that have been pruned previously, if any, via prop. 6. Every single quantum operation is
removed per iteration, ensuring monotonic circuit complexity reduction while preserving dominant
unitary evolution characteristics; see details in Alg. 2. We experiment with different window sizes
$n$, visualizing their performance on combinatorial iterative QNN pruning in Fig. 7-(a,b). Our
experimental results demonstrate that iterative pruning with *QuBS* achieves performance comparable
to its global variant with significantly reduced computational resources: we observe that a QUBO
subproblem dimension of $n = 15$ delivers performance very close to the global *QuBS* baseline $n = 50$

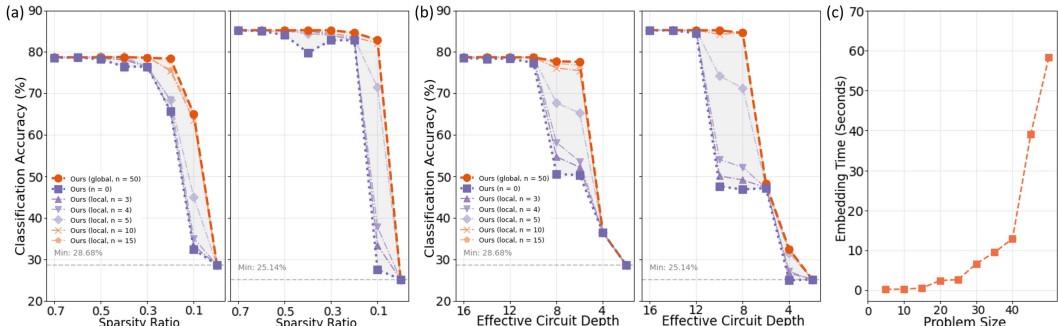

Figure 7: Iterative combinatorial QNN pruning behavior on: (a) gate-level, (b) depth-level, of different window sizes $n$; ours ($n = 0$) means QNN is pruned only based on $\mathcal{P}(\cdot)$. Time required for minor embedding search on the annealer hardware across different problem sizes $N$ is visualized in (c).

for our QNN architecture; it represents a 70 percent reduction in the required qubit resources while maintaining equivalent solution quality. As we increase the sliding window size $n$, the performance of the iterative *QuBS* converges to its global variant as expected. Notably, the window size influences problem embedding time as visualized in Fig. 7-(c); the embedding search is performed with the minorminer algorithm (Zbinden et al., 2020).

## 7   DISCUSSION, FUTURE WORK AND CONCLUSION

We introduced *QuBS*, a framework that bridges quantum annealers with combinatorial QNN pruning. We theoretically established that the non-commutative Hamiltonian generator interactions create classically intractable complex QNN pruning landscapes. Empirically, *QuBS* consistently outperforms existing QNN pruning methods, especially under aggressive sparsity constraints, by a margin of 30 percent in preserved classification accuracy and establishes a new standard. We also show that its iterative variant can reduce the required annealer hardware resources by around 70 percent with negligible performance drop compared to its global variant. *We emphasize that QuBS is solver-agnostic by design; its evaluation is performed with annealers as emerging hardware-efficient QUBO solvers in this work.* In general, we demonstrate that *QuBS* provides an effective framework for combinatorial QNN pruning, facilitating the efficient deployment of QNN on near-term quantum devices. For reproducibility and community benefit, we will release our source code.

**Limitations**. While our *QuBS* framework significantly outperforms existing approaches; see our experiment in Sec. 6.1 and 6.2 in QNN pruning, its current combinatorial formulation is restricted to second-order; incorporating higher-order correlations and exploring efficient problem solvers remains an open question.

**Future Work**. Beyond incorporating higher-order interactions, several promising research avenues remain. Adapting *QuBS* for combinatorial quantum neural architecture search is another compelling direction, with potential impact on automated quantum ansatz design. We hope this work encourages further research in QNN pruning and compression.

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

# Appendix

This appendix serves as a comprehensive supplement to the main paper, systematically expanding upon key theoretical and empirical aspects of the study. We provide a rigorous theoretical foundation for QNN combinatorial pruning up to the second order (Sec. A). Then, we give a detailed derivation of forward-only Hessian computation for the classification cost function (Sec. B) Further theoretical insights are provided through an analysis of the coupling sparsity and its relationship to quantum operation commutativity (Sec. C), as well as complete proofs for propositions 1, 2, 3, 4, 6 (Sec. D, E, F, G and H), which establish the mathematical framework for *QuBS*. The algorithmic protocols for implementing *QuBS* are provided in Sec. I. A benchmark study evaluating the effectiveness of annealers as efficient QUBO solvers and the noise effect of current-generation machines is provided in Sec. J. The logical problem embedded on the physical annealer is visualized in Sec. K; the physical resource requirements are given in Sec. L. Finally, an architecture study is provided in Sec. M for completeness.

## A    SECOND-ORDER COMBINATORIAL QNN ARCHITECTURE PRUNING

Projective measurement introduces inherent non-linearity in QNN output, necessitating exhaustive exploration of exponentially many configurations to identify the analytical optimum sparse architecture. We leverage the second-order approximation to estimate this measurement-induced non-linear fidelity change. We detail the framework and its specific application to evaluating fidelity changes for combinatorial QNN pruning, as used in our primary experiments. For a cost function $\mathcal{L}(\mathbf{w}) \in \mathbb{R}$, where $\mathbf{w} = \{w_1, \cdots, w_N\} \in \mathbb{R}^N$ represents tunable QNN components, its change $\delta\mathcal{L}(\mathbf{w}) \in \mathbb{R}$ resulting from perturbations $\delta\mathbf{w} \in \mathbb{R}^N$ around current parameters $\mathbf{w}_0 \in \mathbb{R}^N$ is given by:

$$\delta\mathcal{L}(\mathbf{w}) = \underbrace{\nabla\mathcal{L}(\mathbf{w}_0)^\top \delta\mathbf{w}}_{\text{First-order term}} + \frac{1}{2}\underbrace{\delta\mathbf{w}^\top \nabla^2\mathcal{L}(\mathbf{w}_0)\delta\mathbf{w}}_{\text{Second-order term}} + \underbrace{\mathcal{O}(\|\delta\mathbf{w}\|^3)}_{\text{Higher-order correlations}}, \tag{16}$$

where

$$\nabla\mathcal{L}(\mathbf{w}_0) = \frac{\partial\mathcal{L}}{\partial\mathbf{w}}(\mathbf{w}_0) \in \mathbb{R}^N, \quad \delta\mathcal{L}(\mathbf{w}) = \mathcal{L}(\mathbf{w}) - \mathcal{L}(\mathbf{w}_0),$$
$$\nabla^2\mathcal{L}(\mathbf{w}_0) = \frac{\partial^2\mathcal{L}}{\partial\mathbf{w}^2}(\mathbf{w_0}) \in \mathbb{R}^{N\times N}, \quad \delta\mathbf{w} = \mathbf{w} - \mathbf{w}_0. \tag{17}$$

$\mathcal{O}$ denotes asymptotic order notation. We reparametrize weights $\mathbf{w}$ as:

$$\mathbf{w} = \mathbf{w}_0 \odot (1 - \mathbf{m}). \tag{18}$$

where $\odot$ denotes the Hadamard product, and $\mathbf{m} \in \{0,1\}^N$ is a binary mask indicating removed quantum operations ($m_i = 1$) or retained ones ($m_i = 0$). The resulting parameter perturbations $\delta\mathbf{w}_{pr} \in \mathbb{R}^N$ are:

$$\delta\mathbf{w}_{pr} = \mathbf{w}_0 \odot (-\mathbf{m}). \tag{19}$$

The optimization objective minimizes the fidelity change, up to the 2nd order, under target sparsity constraints:

$$\operatorname*{arg\,min}_{\mathbf{m}\in\mathbb{B}^N}\left[\nabla\mathcal{L}(\mathbf{w}_0)^\top\delta\mathbf{w}_{pr} + \frac{1}{2}(\delta\mathbf{w}_{pr})^\top\nabla^2\mathcal{L}(\mathbf{w}_0)(\delta\mathbf{w}_{pr})\right], \quad \|\mathbf{m}\|_0 = N_{\text{prune}}. \tag{20}$$

$\|\mathbf{m}\|_0$ denotes the $\ell_0$-norm of the binary mask $\mathbf{m}$, counting the number of removed operations, i.e. where $m_i = 1$. The equivalence between removing a parametrized gate $U_k(w_k)$ and nullifying its parameter $w_k \to 0$ is formally established by the operator-norm convergence $\lim_{\|w_k\|\to 0} U_k(w_k) = I$, where the gate converges to the identity operator. This behavior is a **direct consequence of the Lie group structure underlying quantum gates**. In this framework, a parametrized gate is generated via the exponential map, $U_k(w_k) = e^{iw_k H_k}$, where $H_k$ is a Hermitian generator from the associated Lie algebra. The identity operator $I$ is the canonical identity element of the Lie group. As the parameter $w_k$ approaches zero, the exponential map ensures a smooth and continuous path within the group, guaranteeing that the unitary transformation collapses directly to this identity element, $e^{i\cdot 0\cdot H_k} = I$. Thus, the Lie group structure provides the necessary mathematical rigor, ensuring that parameter nullification is a well-defined and continuous operation that corresponds precisely to the physical act of gate removal.

# B    FORWARD-ONLY COUPLING EVALUATIONS FOR QNNs

**Review: Hessian for the cost function**. Hessian matrix of the cost function $C(\boldsymbol{\theta}; \mathbf{x})$ can be efficiently evaluated with forward-only quantum circuit evaluations (Teo, 2023); the cost function takes the form of:

$$C(\boldsymbol{w}; \mathbf{x}) = \langle 0 | \mathbf{z}^\dagger(\mathbf{x}) U^\dagger(\boldsymbol{w}) \, O \, U(\boldsymbol{w}) \mathbf{z}(\mathbf{x}) | 0 \rangle, \tag{21}$$

where $\mathbf{z}(\mathbf{x})$ is a unitary operation that embeds classical data in quantum states and $U(\boldsymbol{w})$ is a parameterized QNN. $\boldsymbol{w}$ represents trainable parameters of the QNN, and $\mathbf{x}$ represents classical input data. The Hessian $\mathbf{H}$ is defined as:

$$\mathbf{H}_{ij} = \frac{\partial^2 C(\boldsymbol{w}; \mathbf{x})}{\partial w_i \partial w_j}. \tag{22}$$

Starting with the parameter-shift rule for first-order derivatives: for a parameter $w_k$ with generator $G_k$ (satisfying $G_k^2 = I$), the gradient component can be derived from the trigonometric structure as:

$$\frac{\partial C}{\partial w_k} = \frac{C(w_k + s) - C(w_k - s)}{2 \sin s}, \quad \left( s = \frac{\pi}{2} \text{ for Pauli-generated gates} \right). \tag{23}$$

For diagonal Hessian elements $\mathbf{H}_{ii}$, we apply shift rule twice on $\theta_i$:

$$\begin{aligned}
\mathbf{H}_{ii} &= \frac{\partial}{\partial w_i} \left( \frac{\partial C}{\partial w_i} \right) \\
&= \frac{1}{2 \sin s} \frac{\partial}{\partial w_i} \Big[ C(w_i + s) - C(w_i - s) \Big] \\
&= \frac{1}{4 \sin^2 s} \Big[ C(w_i + 2s) - 2C(w_i) + C(w_i - 2s) \Big].
\end{aligned} \tag{24}$$

For off-diagonal elements $\mathbf{H}_{ij}$ ($i \neq j$), we differentiate sequentially in orthogonal directions:

$$\begin{aligned}
\mathbf{H}_{ij} &= \frac{\partial}{\partial w_j} \left( \frac{\partial C}{\partial w_i} \right) \\
&= \frac{1}{2 \sin s} \frac{\partial}{\partial w_j} \Big[ C(w_i + s) - C(w_i - s) \Big] \\
&= \frac{1}{4 \sin^2 s} \Big[ C(w_i + s, w_j + s) - C(w_i + s, w_j - s) \\
&\quad - C(w_i - s, w_j + s) + C(w_i - s, w_j - s) \Big].
\end{aligned} \tag{25}$$

This method yields an exact analytical Hessian, requiring $\mathcal{O}(d^2)$ forward-only circuit evaluations for $d$ parametrized quantum operations. Each evaluation corresponds to a unique parameter combination in $\{w_i \pm s, w_j \pm s\}$, which can be implemented via concurrent adjustments to gate parameters in quantum hardware control systems.

**Remark 4.** *The protocol remains valid even when $\mathbf{x}$ is preprocessed by trainable neural networks under a hybrid quantum-classical learning paradigm.*

**Hessian for our classification problem**. For our classification experiments, we leverage cross-entropy costs using projection operators. Let $\{\Pi_{\mathbf{y}} = |\mathbf{y}\rangle\langle\mathbf{y}|\}$ define a set of projectors onto computational basis states, with measurement probabilities:

$$P(\mathbf{y}|\mathbf{x}) = \langle 0 | \mathbf{z}^\dagger(\mathbf{x}) U^\dagger(\boldsymbol{w}) \Pi_{\mathbf{y}} U(\boldsymbol{w}) \mathbf{z}(\mathbf{x}) | 0 \rangle. \tag{26}$$

The cross-entropy cost function is defined as:

$$C(\boldsymbol{w}; \mathbf{x}) = - \sum_{\mathbf{y}} \mathbf{y}_{\text{true}} \log P(\mathbf{y}|\mathbf{x}), \tag{27}$$

where $\mathbf{y}_{\text{true}}$ is a one-hot encoded label. For parameter $w_k$ with generator $G_k$ ($G_k^2 = I$), we can derive:

$$\frac{\partial C}{\partial w_k} = - \sum_{\mathbf{y}} \mathbf{y}_{\text{true}} \frac{\langle \Pi_{\mathbf{y}}(w_k + s) \rangle - \langle \Pi_{\mathbf{y}}(w_k - s) \rangle}{2 \sin s \cdot P(\mathbf{y}|\mathbf{x})}, \tag{28}$$

where $\langle\Pi_{\mathbf{y}}(w_k \pm s)\rangle$ denotes the projector expectation with shifted parameters. The Hessian entries can then be expressed as:

$$\mathbf{H}_{ij} = \frac{1}{4\sin^2 s} \sum_{\mathbf{y}} \mathbf{y}_{\text{true}} \left[ \frac{\Delta\Pi_{\mathbf{y}}^{(i)} \Delta\Pi_{\mathbf{y}}^{(j)}}{P^2(\mathbf{y}|\mathbf{x})} - \frac{\Box\Pi_{\mathbf{y}}^{(ij)}}{P(\mathbf{y}|\mathbf{x})} \right], \tag{29}$$

where:

$$\Delta\Pi_{\mathbf{y}}^{(k)} = \langle\Pi_{\mathbf{y}}(w_k + s)\rangle - \langle\Pi_{\mathbf{y}}(w_k - s)\rangle, \tag{30}$$

$$\Box\Pi_{\mathbf{y}}^{(ij)} = \langle\Pi_{\mathbf{y}}(w_i+s, w_j+s)\rangle - \langle\Pi_{\mathbf{y}}(w_i+s, w_j-s)\rangle$$
$$-\langle\Pi_{\mathbf{y}}(w_i-s, w_j+s)\rangle + \langle\Pi_{\mathbf{y}}(w_i-s, w_j-s)\rangle. \tag{31}$$

## C  CONNECTIONS BETWEEN COUPLING SPARSITY AND QUANTUM GATE COMMUTABILITY FOR QNN

Consider a general QNN defined by a parametrized unitary $U(\mathbf{w})$ with parameters $\mathbf{w} = (\ldots, w_i, \ldots)$ and input state $|\psi_0\rangle$. The output state is $|\psi(\mathbf{w})\rangle = U(\mathbf{w})|\psi_0\rangle$ and the cost for a Hermitian observable $H$ is

$$J(\mathbf{w}) = \langle\psi(\mathbf{w})|H|\psi(\mathbf{w})\rangle. \tag{32}$$

Define the effective Hermitian generators

$$\mathcal{G}_i(\mathbf{w}) := i\, U^\dagger(\mathbf{w}) \frac{\partial U(\mathbf{w})}{\partial w_i}, \tag{33}$$

so that $U^\dagger\partial_{w_i}U = -i\mathcal{G}_i$. Using $\partial_{w_i}U = -i\,U\mathcal{G}_i$ and $\partial_{w_i}U^\dagger = i\,\mathcal{G}_iU^\dagger$, the product rule gives the first derivative

$$\frac{\partial J}{\partial w_i} = \langle\psi_0|\Big(\frac{\partial U^\dagger}{\partial w_i}\Big)HU|\psi_0\rangle + \langle\psi_0|U^\dagger H\Big(\frac{\partial U}{\partial w_i}\Big)|\psi_0\rangle = i\,\langle\psi(\mathbf{w})|[\widetilde{\mathcal{G}}_i(\mathbf{w}), H]|\psi(\mathbf{w})\rangle, \tag{34}$$

where

$$\widetilde{\mathcal{G}}_i(\mathbf{w}) := U(\mathbf{w})\,\mathcal{G}_i(\mathbf{w})\,U^\dagger(\mathbf{w}). \tag{35}$$

To obtain the Hessian, we differentiate the gradient $\frac{\partial J}{\partial w_i} = i\,\langle\psi(\mathbf{w})|[\widetilde{\mathcal{G}}_i(\mathbf{w}), H]|\psi(\mathbf{w})\rangle$. Letting $A := [\widetilde{\mathcal{G}}_i, H]$, we write $H_{ij} = \partial_{w_j}(i\langle\psi|A|\psi\rangle)$. Applying the product rule yields

$$H_{ij} = i\Big[(\partial_{w_j}\langle\psi|)A|\psi\rangle + \langle\psi|(\partial_{w_j}A)|\psi\rangle + \langle\psi|A(\partial_{w_j}|\psi\rangle)\Big]. \tag{36}$$

To evaluate the state derivatives, note that $\partial_{w_j}U = -iU\mathcal{G}_j$, which implies $\partial_{w_j}|\psi\rangle = -i\,\widetilde{\mathcal{G}}_j|\psi\rangle$ and $\partial_{w_j}\langle\psi| = i\,\langle\psi|\widetilde{\mathcal{G}}_j$. Since $H$ is parameter-independent, the only source of parameter dependence in $A$ is $\widetilde{\mathcal{G}}_i$, giving $\partial_{w_j}A = [\partial_{w_j}\widetilde{\mathcal{G}}_i, H]$. Substituting these three derivative expressions back into the product-rule expansion gives

$$H_{ij} = i\Big[i\langle\psi|\widetilde{\mathcal{G}}_j A|\psi\rangle - i\langle\psi|A\widetilde{\mathcal{G}}_j|\psi\rangle\Big] + i\langle\psi|[\partial_{w_j}\widetilde{\mathcal{G}}_i, H]|\psi\rangle. \tag{37}$$

Recognizing the first two terms as a commutator then yields

$$H_{ij} = -\langle\psi|[\widetilde{\mathcal{G}}_j, A]|\psi\rangle + i\,\langle\psi|[\partial_{w_j}\widetilde{\mathcal{G}}_i, H]|\psi\rangle, \tag{38}$$

and substituting $A = [\widetilde{\mathcal{G}}_i, H]$ gives the exact Hessian:

$$H_{ij}(\mathbf{w}) = -\langle\psi(\mathbf{w})|\,[\widetilde{\mathcal{G}}_j(\mathbf{w}), [\widetilde{\mathcal{G}}_i(\mathbf{w}), H]]\,|\psi(\mathbf{w})\rangle + i\,\langle\psi(\mathbf{w})|\,[\partial_{w_j}\widetilde{\mathcal{G}}_i(\mathbf{w}), H]\,|\psi(\mathbf{w})\rangle. \tag{39}$$

**Remark 5.** *This analysis examines the low-level expectation value cost function of an arbitrary Hermitian observable $H$, making it naturally generalize to any cost function that incorporates QNN outputs, including, but not limited to, the cross-entropy metric employed in this experiment.*

## D    PROOF OF PROPOSITION 2

We provide proof of Prop. 2 as given in the main text:

$$\arg\min_{\boldsymbol{m}\in\mathbb{B}^N} \nabla\mathcal{L}(\boldsymbol{w}_0)^\top \delta\boldsymbol{w}_{\mathrm{pr}} + \frac{1}{2}(\delta\boldsymbol{w}_{\mathrm{pr}})^\top \nabla^2\mathcal{L}(\boldsymbol{w}_0)\delta\boldsymbol{w}_{\mathrm{pr}}$$

$$= \arg\min_{\boldsymbol{m}\in\mathbb{B}^N} \nabla\mathcal{L}(\boldsymbol{w}_0)^\top \boldsymbol{w}_0\odot(-\boldsymbol{m}) + \frac{1}{2}(\boldsymbol{w}_0\odot(-\boldsymbol{m}))^\top \nabla^2\mathcal{L}(\boldsymbol{w}_0)(\boldsymbol{w}_0\odot(-\boldsymbol{m}))$$

$$= \arg\min_{\boldsymbol{m}\in\mathbb{B}^N} \nabla\mathcal{L}(\boldsymbol{w}_0)^\top \operatorname{diag}(\boldsymbol{w}_0)(-\boldsymbol{m}) + \frac{1}{2}(\operatorname{diag}(\boldsymbol{w}_0)(-\boldsymbol{m}))^\top \nabla^2\mathcal{L}(\boldsymbol{w}_0)(\operatorname{diag}(\boldsymbol{w}_0)(-\boldsymbol{m})) \quad (40)$$

$$= \arg\min_{\boldsymbol{m}\in\mathbb{B}^N} -\nabla\mathcal{L}(\boldsymbol{w}_0)^\top \operatorname{diag}(\boldsymbol{w}_0)\boldsymbol{m} + \frac{1}{2}\boldsymbol{m}^\top \operatorname{diag}(\boldsymbol{w}_0)^\top \nabla^2\mathcal{L}(\boldsymbol{w}_0)\operatorname{diag}(\boldsymbol{w}_0)\boldsymbol{m}$$

$$= \arg\min_{\boldsymbol{m}\in\mathbb{B}^N} \boldsymbol{c}^\top\boldsymbol{m} + \boldsymbol{m}^\top\boldsymbol{Q}\boldsymbol{m}.$$

Via careful inspection and performing variable substitution, we have:

$$\boldsymbol{c}^\top = -\nabla\mathcal{L}(\boldsymbol{w}_0)^\top\operatorname{diag}(\boldsymbol{w}_0) = -\begin{bmatrix}\cdots & \frac{\partial\mathcal{L}}{\partial w^i}(\boldsymbol{w}_0) & \cdots\end{bmatrix}\begin{bmatrix}\ddots & & \boldsymbol{0} \\ & w_0^i & \\ \boldsymbol{0} & & \ddots\end{bmatrix} = -\begin{bmatrix}\cdots & w_0^i\frac{\partial\mathcal{L}}{\partial w^i}(\boldsymbol{w}_0) & \cdots\end{bmatrix},$$

$$(41)$$

$$\boldsymbol{Q} = \frac{1}{2}\operatorname{diag}(\boldsymbol{w}_0)\nabla^2\mathcal{L}(\boldsymbol{w}_0)\operatorname{diag}(\boldsymbol{w}_0) = \frac{1}{2}\begin{bmatrix}(w_0^1)^2\frac{\partial^2\mathcal{L}}{\partial(w^1)^2}(\boldsymbol{w}_0) & \cdots & w_0^1 w_0^N\frac{\partial^2\mathcal{L}}{\partial w^1\partial w^N}(\boldsymbol{w}_0) \\ \vdots & \ddots & \vdots \\ w_0^N w_0^1\frac{\partial^2\mathcal{L}}{\partial w^N\partial w^1}(\boldsymbol{w}_0) & \cdots & (w_0^N)^2\frac{\partial^2\mathcal{L}}{\partial(w^N)^2}(\boldsymbol{w}_0)\end{bmatrix}.$$

$$(42)$$

## E    PROOF OF PROPOSITION 3

We provide proof of Prop. 3 as given in the main text:

$$\arg\min_{\boldsymbol{m}\in\mathbb{B}^N} \nabla\mathcal{L}(\boldsymbol{w}_0)^\top \delta\boldsymbol{w}_{\mathrm{pr}} + \frac{1}{2}(\delta\boldsymbol{w}_{\mathrm{pr}})^\top \nabla^2\mathcal{L}(\boldsymbol{w}_0)\delta\boldsymbol{w}_{\mathrm{pr}}$$

$$= \arg\min_{\boldsymbol{m}\in\mathbb{B}^N} \boldsymbol{c}^\top\boldsymbol{m} + \boldsymbol{m}^\top\boldsymbol{Q}\boldsymbol{m}$$

$$= \arg\min_{\boldsymbol{m}\in\mathbb{B}^N} \begin{bmatrix}c_1 & \cdots & c_n & \cdots\end{bmatrix}\begin{bmatrix}m_1 \\ \vdots \\ m_n \\ \vdots\end{bmatrix} + \begin{bmatrix}m_1 & \cdots & m_n & \cdots\end{bmatrix}\begin{bmatrix}Q_{11} & \cdots & Q_{1n} & \cdots \\ \vdots & \ddots & \vdots & \ddots \\ Q_{n1} & \cdots & Q_{nn} & \cdots \\ \vdots & \ddots & \vdots & \ddots\end{bmatrix}\begin{bmatrix}m_1 \\ \vdots \\ m_n \\ \vdots\end{bmatrix}.$$

$$(43)$$

For depth-level combinatorial QNN pruning, we enforce quantum operations belonging to the same circuit depth to share the same decision variable such that $m_i = m_{\mathcal{S}_j}$:

$$\arg\min_{\boldsymbol{m}\in\mathbb{B}^N} \begin{bmatrix}c_1 & \cdots & c_n & \cdots\end{bmatrix}\begin{bmatrix}m_{\mathcal{S}_j} \\ \vdots \\ m_{\mathcal{S}_j} \\ \vdots\end{bmatrix} + \begin{bmatrix}m_{\mathcal{S}_j} & \cdots & m_{\mathcal{S}_j} & \cdots\end{bmatrix}\begin{bmatrix}Q_{11} & \cdots & Q_{1n} & \cdots \\ \vdots & \ddots & \vdots & \ddots \\ Q_{n1} & \cdots & Q_{nn} & \cdots \\ \vdots & \ddots & \vdots & \ddots\end{bmatrix}\begin{bmatrix}m_{\mathcal{S}_j} \\ \vdots \\ m_{\mathcal{S}_j} \\ \vdots\end{bmatrix}$$

$$= \arg\min_{\boldsymbol{m}\in\mathbb{B}^M} \begin{bmatrix}\sum_{i=1}^n c_i & \cdots\end{bmatrix}\begin{bmatrix}m_{\mathcal{S}_j} \\ \vdots\end{bmatrix} + \begin{bmatrix}m_{\mathcal{S}_j} & \cdots\end{bmatrix}\begin{bmatrix}\sum_{i,j=1}^n Q_{ij} & \cdots \\ \vdots & \ddots\end{bmatrix}\begin{bmatrix}m_{\mathcal{S}_j} \\ \vdots\end{bmatrix}.$$

$$(44)$$

## F    PROOF OF PROPOSITION 4

We demonstrate how target sparsity constraints, expressed as linear equality constraints, can be incorporated into the QUBO formulation from Prop. 2. We show the proof below:

$$
\begin{aligned}
&\arg\min_{\boldsymbol{m}\in\mathbb{B}^N} \left(\boldsymbol{c}^\top\boldsymbol{m} + \boldsymbol{m}^\top\boldsymbol{Q}\boldsymbol{m}\right) + \lambda(\boldsymbol{A}^\top\boldsymbol{m} - b)^2\\
&= \arg\min_{\boldsymbol{m}\in\mathbb{B}^N} \left(\boldsymbol{c}^\top\boldsymbol{m} + \boldsymbol{m}^\top\boldsymbol{Q}\boldsymbol{m}\right) + \lambda(\boldsymbol{A}^\top\boldsymbol{m} - b)^\top(\boldsymbol{A}^\top\boldsymbol{m} - b)\\
&= \arg\min_{\boldsymbol{m}\in\mathbb{B}^N} \left(\boldsymbol{c}^\top\boldsymbol{m} + \boldsymbol{m}^\top\boldsymbol{Q}\boldsymbol{m}\right) + \lambda(\boldsymbol{m}^\top\boldsymbol{A} - b^\top)(\boldsymbol{A}^\top\boldsymbol{m} - b)\\
&= \arg\min_{\boldsymbol{m}\in\mathbb{B}^N} \left(\boldsymbol{c}^\top\boldsymbol{m} + \boldsymbol{m}^\top\boldsymbol{Q}\boldsymbol{m}\right) + \lambda(\boldsymbol{m}^\top\boldsymbol{A}\boldsymbol{A}^\top\boldsymbol{m} - 2b\boldsymbol{A}^\top\boldsymbol{m} + b^2)\\
&= \arg\min_{\boldsymbol{m}\in\mathbb{B}^N} \boldsymbol{m}^\top(\boldsymbol{Q} + \lambda\boldsymbol{A}\boldsymbol{A}^\top)\boldsymbol{m} + (\boldsymbol{c}^\top - 2b\lambda\boldsymbol{A}^\top)\boldsymbol{m}.
\end{aligned}
\tag{45}
$$

## G    PROOF OF PROPOSITION 5

Assume $\mathbf{q}\in\mathbb{R}^N$ is an eigenvector of $\mathbf{Q}\in\mathbb{R}^{N\times N}$ with eigenvalue $\mu$, the following holds in general for arbitrary matrix $\mathbf{Q}$ and vector $\mathbf{A}\in\mathbb{R}^{N\times 1}$:

$$
\begin{aligned}
&(\mathbf{Q} + \lambda\mathbf{A}\mathbf{A}^\top)\mathbf{q}\\
&= \mu\mathbf{q} + \lambda\mathbf{A}\mathbf{A}^\top\mathbf{q}.
\end{aligned}
\tag{46}
$$

By decomposing the eigenvector $\mathbf{q}$ of $\mathbf{Q}$ into components parallel and orthogonal to $\mathbf{A}$, we have:

$$
\mathbf{q} = \mathbf{q}_{\|} + \mathbf{q}_{\perp}, \quad \text{where } \mathbf{q}_{\|} = c\mathbf{A} = \frac{\mathbf{A}^\top\mathbf{q}}{\mathbf{A}^\top\mathbf{A}}\mathbf{A}, \quad \mathbf{q}_{\perp} = \mathbf{q} - \mathbf{q}_{\|}.
\tag{47}
$$

Substituting the decomposition of $\mathbf{q}$ into Eqn. 46 and due to orthogonality between $\mathbf{q}_{\perp}\in^{N\times 1}$ and $\mathbf{A}$ (i.e. $\mathbf{A}^\top\mathbf{q}_{\perp} = 0$), we have:

$$
\begin{aligned}
&\mu\mathbf{q} + \lambda\mathbf{A}\mathbf{A}^\top(\mathbf{q}_{\|} + \mathbf{q}_{\perp})\\
&= \mu\mathbf{q} + \lambda\mathbf{A}\mathbf{A}^\top\mathbf{q}_{\|}.
\end{aligned}
\tag{48}
$$

Specifically, as $\mathbf{A} = [\cdots, 1, \cdots]^\top$, we have $\mathbf{A}\mathbf{A}^\top = \mathbf{1}\in\mathbb{R}^{N\times N}$, a special rank-1 matrix having only one non-zero eigenvalue: $N$, with the corresponding eigenvector $\mathbf{A}$. Therefore, we have:

$$
\begin{aligned}
&\mu\mathbf{q} + \lambda\mathbf{A}\mathbf{A}^\top\mathbf{q}_{\|}\\
&= \mu\mathbf{q} + \lambda\mathbf{A}\mathbf{A}^\top c\mathbf{A}\\
&= \mu\mathbf{q} + \lambda N c\mathbf{A}\\
&= \mu\mathbf{q} + \lambda N \frac{\mathbf{A}^\top\mathbf{q}}{\mathbf{A}^\top\mathbf{A}}\mathbf{A}.
\end{aligned}
\tag{49}
$$

The perturbed eigenvalue, $\mu'$, is determined by factoring out $\mathbf{q}$:

$$
\begin{aligned}
\mu'\mathbf{q} &= \mu\mathbf{q} + \lambda N \frac{\mathbf{A}^\top\mathbf{q}}{\mathbf{A}^\top\mathbf{A}}\mathbf{A},\\
\mathbf{q}^\top\mu'\mathbf{q} &= \mathbf{q}^\top\mu\mathbf{q} + \lambda N \frac{\mathbf{A}^\top\mathbf{q}}{\mathbf{A}^\top\mathbf{A}}\mathbf{q}^\top\mathbf{A},\\
\mathbf{q}^\top\mu'\mathbf{q} &= \mathbf{q}^\top\mu\mathbf{q} + \lambda N \frac{(\mathbf{A}^\top\mathbf{q})^2}{\mathbf{A}^\top\mathbf{A}}.
\end{aligned}
\tag{50}
$$

Dividing by $\mathbf{q}^\top\mathbf{q}$ gives the effective eigenvalue:

$$
\mu' = \mu + \lambda N \frac{|\langle\mathbf{A}, \mathbf{q}\rangle|^2}{\|\mathbf{A}\|^2\|\mathbf{q}\|^2}.
\tag{51}
$$

Results demonstrate rank-1 perturbation affects the eigenvalue based on the alignment between $\mathbf{q}$ and $\mathbf{A}$, while leaving the orthogonal component $\mathbf{q}_{\perp}$ unchanged.

## H    PROOF OF PROPOSITION 6

We propose a combinatorial iterative QNN pruning methodology that sequentially removes subsets of quantum operations of a QNN while preserving the information of previously removed QNN structures. This approach accounts for cross-term interactions between frozen (previously removed) and active quantum operations, ensuring that the problem at each iteration incorporates the cumulative influence of all prior decisions without losing global information. Consider a sliding window on quantum operations indexed from $n$ to $N$, with frozen quantum operations indexed from 1 to $n - 1$, Eqn. 8 can be reformulated accordingly as:

$$
\underset{\boldsymbol{m} \in \mathbb{B}^{N-n}}{\arg\min} \left( \boldsymbol{c}^\top \boldsymbol{m} + \boldsymbol{m}^\top \boldsymbol{Q} \boldsymbol{m} \right)
$$

$$
= \underset{\boldsymbol{m} \in \mathbb{B}^{N-n}}{\arg\min} \sum_{i=1}^{N} c_i m_i + \sum_{i,j=1}^{N} m_i Q_{ij} m_j. \tag{52}
$$

Decomposing the sums into frozen (1 to $n - 1$) and active ($n$ to $N$) components:

$$
= \underset{\boldsymbol{m} \in \mathbb{B}^{N-n}}{\arg\min} \sum_{i=1}^{n} c_i m_i + \sum_{i=n}^{N} c_i m_i + \sum_{i,j=1}^{n} m_i Q_{ij} m_j
$$

$$
+ \sum_{i=1}^{n}\sum_{j=n}^{N} m_i Q_{ij} m_j + \sum_{i=n}^{N}\sum_{j=1}^{n} m_i Q_{ij} m_j + \sum_{i,j=n}^{N} m_i Q_{ij} m_j. \tag{53}
$$

Eliminating the frozen terms from the objective, we have:

$$
= \underset{\boldsymbol{m} \in \mathbb{B}^{N-n}}{\arg\min} \sum_{i=n}^{N} c_i m_i + \sum_{i=1}^{n}\sum_{j=n}^{N} m_i Q_{ij} m_j
$$

$$
+ \sum_{i=n}^{N}\sum_{j=1}^{n} m_i Q_{ij} m_j + \sum_{i,j=n}^{N} m_i Q_{ij} m_j
$$

$$
= \underset{\boldsymbol{m} \in \mathbb{B}^{N-n}}{\arg\min} \sum_{i=n}^{N} c_i m_i + 2\sum_{i=1}^{n}\sum_{j=n}^{N} m_i Q_{ij} m_j + \sum_{i,j=n}^{N} m_i Q_{ij} m_j \tag{54}
$$

$$
= \underset{\boldsymbol{m} \in \mathbb{B}^{N-n}}{\arg\min} \left[ c_n + 2\sum_{i=1}^{n} m_i Q_{in} \quad \cdots \quad c_N + 2\sum_{i=1}^{n} m_i Q_{iN} \right] \begin{bmatrix} m_n \\ \vdots \\ m_N \end{bmatrix}
$$

$$
+ \begin{bmatrix} m_n & \cdots & m_N \end{bmatrix} \begin{bmatrix} Q_{nn} & \cdots & Q_{nN} \\ \vdots & \ddots & \vdots \\ Q_{Nn} & \cdots & Q_{NN} \end{bmatrix} \begin{bmatrix} m_n \\ \vdots \\ m_N \end{bmatrix}.
$$

This formulation demonstrates that the combinatorial iterative QNN pruning approach preserves global information by incorporating the cross-term interactions between frozen, and active variables in the sliding window, through the updated coupling and bias, while maintaining the quadratic structure within the active window.

## I    ALGORITHMIC PROTOCOL FOR *QuBS*

We present the complete high-level pseudocode for implementing *QuBS* framework, encompassing both gate-level and depth-level QNN structure removal procedures. The pseudocodes for *QuBS* are given in Alg. 1 (global approach) and Alg. 2 (local, sliding window approach).

## J    BENCHMARK STUDY OF *QuBS*

Quantum annealers represent promising platforms for efficiently solving QUBO problems (Kim et al., 2025) and potentially delivering high-quality solutions (Das & Chakrabarti, 2005; Denchev

---

**Algorithm 1** Global *QuBS*

---

**Require:**
- Pretrained QNN $\mathcal{N}$ with quantum operations parametrized by $\mathbf{w} \in \mathbb{R}^N$
- Target sparsity $S_t \in [0, 1)$
- Boolean flag `structured` for depth-wise structural removal

1: **if** `structured = True` **then**
2:      Identify depth-wise QNN structures with index sets $I_k$; see Sec. 5.2.
3: **end if**
4: Formulate QUBO problem:

$$\mathbf{m}^* = \arg\min_{\mathbf{m}} \mathbf{m}^T \mathbf{Q} \mathbf{m} + \mathbf{c}^T \mathbf{m}$$

                               $\triangleright$ See Props. 2–4 for details

5: Solve for $\mathbf{m}^*$ leveraging quantum annealers as efficient QUBO solvers
6: Apply QNN structure removal (gate-/depth-wise): $\mathbf{w} \leftarrow \mathbf{w} \odot (\mathbf{1} - \mathbf{m}^*)$

7: **return** A sparse QNN $\mathcal{N}_p$

---

**Algorithm 2** Iterative Local *QuBS* Guided by Evolution Strength

---

**Require:**
- Pretrained QNN $\mathcal{N}$ with quantum operations parametrized by $\mathbf{w} \in \mathbb{R}^N$
- Target sparsity $S_t \in [0, 1)$
- $n \leq N$ for sliding window size

1: Initialize $k \leftarrow 0$, rank quantum operations by its evolution strength: $\tilde{\mathbf{w}} \leftarrow \text{argsort}(|\mathbf{w}|)$
2: **while** $\frac{k}{N} < S_t$ **do**
3:      Choose the QNN operations under the sliding window
4:      Account for previously removed QNN structures and formulate reduced QUBO problem:

$$\mathbf{m}^* = \arg\min_{\mathbf{m}} \mathbf{m}^T \mathbf{Q} \mathbf{m} + \mathbf{c}^T \mathbf{m}$$

                               $\triangleright$ See Props. 2, 3, and 5 for details

5:      Solve for $\mathbf{m}^*$ leveraging quantum annealers as efficient QUBO solvers
6:      Apply QNN structure removal: $\mathbf{w} \leftarrow \mathbf{w} \odot (\mathbf{1} - \mathbf{m}^*)$
7:      Update counter $k \leftarrow k + 1$
8: **end while**

9: **return** A sparse QNN $\mathcal{N}_p$

---

et al., 2016); however, the solution quality can be compromised by various noise sources in current-generation hardware. To establish a baseline for evaluating *QuBS* on combinatorial pruning of QNN with available D-Wave annealers, we: 1) perform an empirical comparison between noisy quantum annealing and classical noiseless simulated annealing-an established method natively provided by D-Wave's software tools for benchmarking quantum annealer's performance and quantifying hardware noise effects; and 2) investigate into how effectively the second-order combinatorial approximation approaches the ground-truth solution via exhaustive search, which guarantees global optimality but soon becomes computationally intractable due to exponential scaling, i.e. $2^N$, of the search space. For this benchmark study, we maintain the same QNN structure as in Fig. 4 but reduce the circuit scale to $M = 2$ qubits to accommodate exhaustive search. To evaluate *QuBS*'s ability to distinguish between functionally relevant and irrelevant quantum operations and assess its robustness against over-parameterization, we also incorporate entangled parameterized phase gates that do not contribute to the cost function (defined as Pauli-Z measurement). These phase gates serve as structural noise in our benchmark study, representing non-functional parameterized operations that may occur in practical QNN designs; this design ensures our validation reflects real-world conditions where quantum circuits may contain unnecessary complexity. The QNN under benchmark has $N = 14$ parametrized quantum gates; see Fig. 8 for the schematic structure of the QNN under benchmark.

**Remark 6.** *Note that simulated annealing exhibits exponential time complexity in the worst case for solving NP-hard QUBO problems (Kim et al., 2025).*

We perform an exhaustive search on the QNN under benchmark across all pruning mask configurations and compare with its second-order combinatorial approximations; see Fig. 8. Two key observations include: 1) the discretized cost landscape exhibits non-convex topology, evidenced by non-monotonic cost function trajectories during progressive pruning of target QNN; 2) second-order combinatorial approximation agrees well with the numerical ground truth derived via exhaustive search, validating the theoretical foundation of *QuBS*. We then benchmark the performance of quantum annealers against their simulated noiseless counterpart to evaluate the noise effect across different sparsity levels. Two key metrics are used: 1) Hamming distance to the ground truth derived from exhaustive search, and 2) energy of the returned solution. Results are recorded in Tab. 3. Empirical results indicate that both using the quantum annealer and simulated annealing can identify solutions that closely approximate—and sometimes identify—the optimal configuration. While quantum

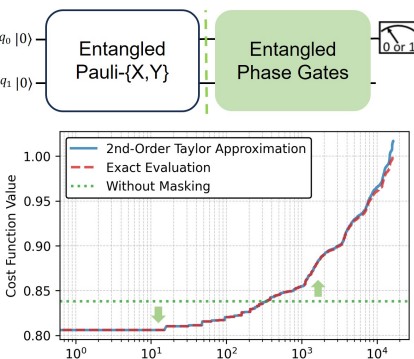

Figure 8: (Top): QNN architecture under benchmark; (bottom): 2nd order combinatorial approximation performance; the green dotted line represents the original QNN without any component removal.

annealers are efficient solvers for QUBO problems, the presence of noise in current-generation hardware can introduce perturbations that reduce solution quality, particularly when the energy gap between neighboring excited states is narrow (Kato, 1950); this is reflected in Tab. 3. The results in the primary experimental section already incorporate these noise effects, demonstrating the practical utility of available quantum annealers despite the discussed limitations.

## K MINOR EMBEDDING VISUALIZATION ON ANNEALER'S TOPOLOGY

The embedding process connects physical qubits into chains to represent a single logical variable, thereby overcoming the limited connectivity of the hardware graph. Fig. 9 provides a visualization of the hardware embeddings on the D-Wave Advantage Series QPU of our logical QUBO problem designed for combinatorial QNN pruning. Specifically, it visualizes how the abstract logical problem graph—where nodes represent binary variables and edges represent quadratic couplings—is mapped onto the physical Pegasus topology of the annealer. Compared to the previous Chimera architectural topology, the enhanced connectivity of the Pegasus topology facilitates

| Metric | Method | Nr. of Optimized Quantum Gates ($N_p$) / $N_{total}$ | | | | |
| --- | --- | --- | --- | --- | --- | --- |
| | | 1/14 | 4/14 | 7/14 | 10/14 | 13/14 |
| H.D. | SA | 0 | 0 | 0 | 0 | 0 |
| | QA | 0 | 0 | 0 | 0 | 2 |
| Energy | SA | -0.0226 | -0.0322 | -0.0322 | 0.0112 | 0.1480 |
| | QA | -0.0226 | -0.0322 | -0.0322 | 0.0112 | 0.1482 |

Table 3: Performance metrics ("H.D." denotes Hamming distance). Performance differences are underscored for emphasis.

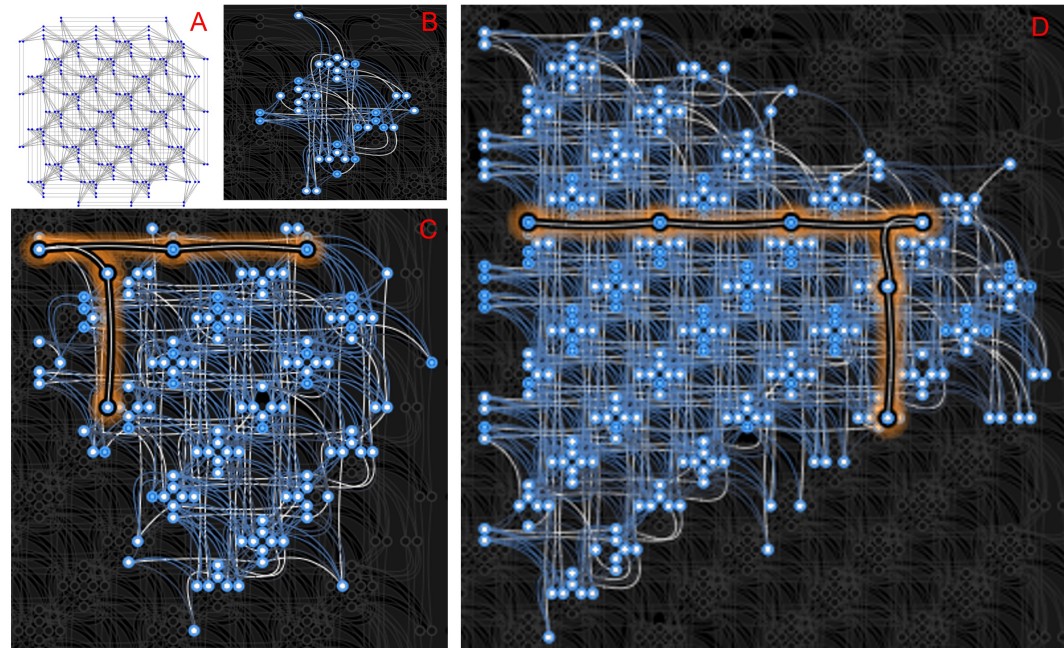

Figure 9: (A) The Pegasus topology structure; (B)–(D) visualization of embeddings on QPUs for test scenarios involving $N = 10$, $N = 20$, and $N = 30$ quantum operations to consider, respectively. In each case, the longest chain within the embedding is highlighted.

more sophisticated embedding patterns while mitigating the need for long chains, demonstrating a measurable improvement in embedding efficiency for large-scale problems.

## L  PHYSICAL RESOURCE ANALYSIS FOR MINOR EMBEDDING

Our analysis in Fig. 10 quantifies the physical resource overhead for embedding logical problems onto QPUs of quantum annealers, evaluating two critical metrics: 1) the number of physical qubits required, and 2) the embedding complexity characterized by the maximum chain length under hardware-native connectivity constraints. The former dictates scalability under finite qubit counts, while the latter reflects the challenges of mapping logical interactions to sparse physical architectures. Examination across problem scales reveals that as logical problem scales linearly, physical qubit demands exhibit piecewise-linear scaling with monotonically increasing slopes, driven by embedding constraints (e.g., matching logical interactions to sparse physical couplers) and error mitigation overhead. Concurrently, the embedding process grows progressively complex, requiring longer chains to preserve logical connectivity while elongated chains amplify susceptibility to chain breaks during annealing, degrading computational fidelity. Nevertheless, we show that quantum annealers can give high-quality solutions for combinatorial QNN pruning as demonstrated, while ongoing hardware architectural improvements would continue to expand the boundaries of embeddable problem sizes and enhance the solution quality.

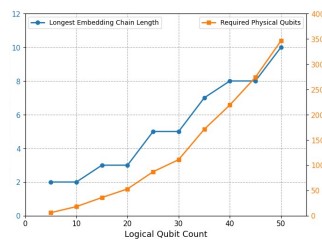

Figure 10: Physical resources for minor embedding.

## M  ARCHITECTURE STUDY

For completeness, we perform additional experiments evaluating the classical backbone feature extractor within the QNN architecture described in Fig. 2 by masking out the entire quantum circuit while maintaining identical experimental setups. The performance comparison, visualized in

Fig. 11, yields three principal observations: (1) the original (without mask) QNN model exhibits suboptimal initialization performance, which is expected given our primary focus on combinatorial QNN pruning rather than architectural optimality—particularly regarding the initial Hilbert space representation of images, as noted in remark 3; (2) the original QNN demonstrates smoother training dynamics throughout the training process despite identical learning rate schedules, suggesting that the quantum evolution potentially introduces regularization effects that stabilize gradient flow; and (3) both architectures eventually converge to comparable asymptotic performance levels, indicating that while the quantum evolution affects learning dynamics, it does not fundamentally alter final representational capacity under these controlled conditions.

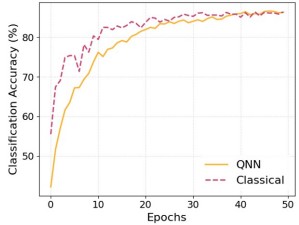

Figure 11: Architecture study.

## N  EXPERIMENTS ON A QUANTUM ANNEALER WITH 10-QUBIT-CIRCUIT

We extended the experiments from Fig. 5 to a 10-qubit circuit. The original depth-2 ansatz, however, yields a QUBO problem requiring 150 logical qubits—a scale that exceeds the capacity of current-generation annealers on which we perform experiments. Consequently, we reduce the circuit depth to 1 to maintain experimental tractability while still enabling a meaningful evaluation. The performance on this 10-qubit configuration, detailed in Table 4, demonstrates that QuBS retains its efficacy and outperforms competing methods by a significant margin.

| Method | Epoch | Sparsity | | | | | | | | |
|---|---|---|---|---|---|---|---|---|---|---|
| | | 1.00 | 0.70 | 0.60 | 0.50 | 0.40 | 0.30 | 0.20 | 0.10 | 0.00 |
| QuBS (ours) | 15 | 80.20 | 80.80 | 80.50 | 80.30 | 79.70 | 79.50 | 79.30 | 62.70 | 45.20 |
| | 35 | **87.00** | **86.20** | **86.10** | **85.30** | **85.10** | **84.70** | **84.20** | **83.50** | **47.75** |
| NR-QNN | 15 | 80.20 | 80.80 | 80.20 | 77.10 | 73.50 | 72.20 | 61.80 | 49.50 | 45.20 |
| | 35 | **87.00** | **86.20** | **86.10** | 82.40 | 80.10 | 72.70 | 69.80 | 51.30 | **47.75** |
| QAdaPrune | 15 | 80.20 | 80.50 | 75.80 | 68.60 | 64.20 | 66.40 | 53.40 | 47.70 | 45.20 |
| | 35 | **87.00** | **86.20** | 85.20 | 70.20 | 62.10 | 58.50 | 57.20 | 48.70 | **47.75** |

Table 4: Top-1 preserved classification accuracy comparison on a 10-qubit circuit across different training epochs. Best values in each sparsity column are bolded.

