# OpenReview forum: "QuBS: Combinatorial Quantum Brain Surgeon on Quantum Annealers"
_ICLR.cc/2026/Conference — ICLR 2026 Conference Desk Rejected Submission_

### Official Review · Reviewer_iudq · 2025-10-22

**Soundness:** 2
**Presentation:** 2
**Contribution:** 2
**Rating:** 2
**Confidence:** 4

**Summary:**

The paper proposes a novel framework, called Combinatorial Quantum Brain Surgeon, to prune QNNs. They benchmark their approach against state-of-the-art on small, feasible instances, showcasing improvement over them. Further, they utilize QUBO problems that can be solved on quantum annealers.

While the approach is interesting, I do think that there is an inconsistency in the problem setting, in particular, one of their primary motivations of pruning QNNs is to avoid trainability issues due to excessive randomness in the models (2-designs), however, pruning takes place after training. The method aims to minimize the increased loss resulting from removal of gates, thus, it is a precondition to work with an already trained model. This would leave the only reason to do QNN pruning to reducing decoherence and operational errors, which already occur during training anyway (i.e., I train my model on a noisy loss, so ideally the model is already trained to be robust against that). That leaves it an open question to me about what the whole motivation of doing QNN pruning is. In classical ML this is usually done afterward to minimize HW requirements, energy, etc., but this seems to not translate into (today's) QML.

It is not clear what the difference between Figure 3 and Appendix P (which should be referenced in the main text where appropriate) is, however, Appendix P shows that the proposed architecture (classical feature extraction + QNN) works better (or for later epochs equally well) when removing the quantum components, which raises the question about the appropriateness of using said architecture. If adding a QNN to the problem setup does not lead to an increase in performance, it seems pointless to even add it.

Further, I think the paper has structural problems. A lot of text has been moved to the Appendix, sometimes, even without reference where necessary (i.e., introduction to adiabatic quantum computing, QUBO) and there are missing links to literature. I want to highlight that the Limitation section should definitely not be hidden (for sure not without even a reference) in the Appendix, as it is a fundamental part of the approach. I think the main paper could at least outline proof sketches, not outsource everything to the Appendix (it should be possible to follow the propositions etc. only from reading the paper) and the proofs in the Appendix could benefit from explanations and elaborations.

**Strengths:**

- The authors have equipped their approach with a sound theoretical framework
- Comparison to state-of-the-art shows an improvement

**Weaknesses:**

- Their motivation for QNN pruning is inconsistent with the approach (outlined above)
- The experimental setup is not clear to me (see comment above)
- Missing comments and explanations for the proofs
- Claims that are not justified: i.e., "Euclidean-space heuristics are misaligned with the dynamics of quantum circuit". This is highlighted many times, however, a justification, in particular, for the relevance to QNN pruning is never given.
- Notation inconsistency. N is sometimes used for qubits, other times for the number of gates. The interaction graph and chromatic number are hardly explained. Eq.13 is not clear (what is $N_p$ and $b$?). App.C. why is $\delta L(w)$ a real number? $\delta$ is a vector of perturbations for every parameter, no? Shouldn't it be $L(\delta w)$? Please restate the propositions in the appendix before proving them and comment/explain on the transformations being done, this would greatly improve readability (also the proofs in matrix form are hardly possible to follow).  Adjust indices in Eq.56 - mathematical notation usually includes lower and upper bound.
- Please add the performance of the full model to the plots comparing the pruning approaches
- The paper does not add evidence to the existence of lottery tickets in QNNs. Either discuss and provide evidence in the main part of the paper, but this is a strong claim to add to the conclusion.
- Add references to the AQC and QUBO introductions
- It is a bit difficult for me to understand why the authors made the decision to only give an introduction to QNNs (from a very basic perspective) in the main part of the paper, and not do the same for QUBO and AQC. I'd suggest giving a very brief introduction there as well, or at least, link to the Appendix. Further, I think the introduction to QNNs does not clearly draw a distinction between their setup and the general definition, i.e., QNNs (at least hardware-efficient) often do not use CR operations (your setup), but work with CNOT, CZ. Further, other types of feature encoding (e.g., re-uploading) can be used too. Lastly, the last sentence in Section 3 needs elaboration, it is not clear to me what this should mean.
- (minor) The cost function and expectation value are used interchangeably, it should be made clear that this is not the case (in particular, if you're training against ground truth labels).

**Questions:**

- Why is the removal and parameter nullification protocol a result of the Lie group framework? This just follows from not rotating being equal to identity.
- What does remark 4 mean?
- Where does the 1/2 factor in Eq.46 go?
- What is the difference between Appendix P and Figure 3?
- Figure 8: What does this figure depict? What is the x-axis, and why is the discrepancy between original and pruned networks so big, even for hardly any components removed?

---

> ### Author Response · Authors · 2025-11-28
>
> ***(Q1): Their motivation for QNN pruning is inconsistent with the approach:*** It is correct that our pruning occurs post-training, and thus does not directly address trainability issues like barren plateaus during the optimization process. However, the reference to 2-design behavior serves to motivate why sparse architectures are desirable in general—not only for trainability but also for their inherent robustness and lower operational overhead. We also agree that one important motivation is to enhance robustness on near-term hardware by reducing susceptibility to decoherence and cumulative errors.
>
> ***Regarding hardware-aware training:*** while training on noisy hardware can impart some robustness, it does not eliminate the benefits of sparsity. For example, sparser circuits: 1) reduce execution time, thereby decreasing the window for decoherence; 2) enable deployment on more constrained devices; 3) lower the cumulative effect of gate errors, even on pre-characterized hardware. These advantages persist regardless of whether the model was trained with noise awareness. Finally, developing pruning methodologies now establishes a crucial foundation for future scalability. As quantum hardware matures and QNNs grow in complexity, the ability to compress models without sacrificing performance will become increasingly critical—much like in classical ML.
>
> ***(Q2): The experimental setup is not clear to me:*** Hybrid quantum-classical (quantum circuits + classical neural net) models are commonly employed in quantum machine learning, particularly for handling high-dimensional classical data such as images; see works “Quantum Machine Learning for Image Classification” [Senokosov et al, 2023]. While establishing clear quantum advantage for such models on real-world tasks remains an active research area, this does not affect the validity of using them as testbeds for developing and evaluating pruning methodologies. Our work focuses specifically on establishing an effective pruning framework that can be applied to pre-trained parametrized quantum circuits, regardless of the ultimate comparative advantage of hybrid models over classical counterparts.
>
> ***(Q3): Missing comments and explanations for the proofs:*** The proposition proofs serve as a complement and therefore do not impact the logical flow of the manuscript. We also carefully revised the manuscript to improve readability.
>
> ***(Q4): Claims that are not justified:*** i.e., “Euclidean-space heuristics are misaligned with the dynamics of quantum circuit”. This is highlighted many times, however, a justification, in particular, for the relevance to QNN pruning is never given: we kindly ask the reviewer to refer to our detailed response to Q3 from Reviewer y9p5.
>
> ***(Q5): Notation inconsistency:*** we thank the reviewers for providing suggestions to improve the manuscript. We performed a thorough revision of the manuscript to ensure all notations are standardized, clearly defined, and used consistently across all sections and equations.
>
> ***(Q6): Please add the performance of the full model to the plots comparing the pruning approaches:*** the complete training dynamics and final performance metrics of the full, unpruned model are provided in the supplementary material (see Figure 11).
>
> ***(Q7): The paper does not add evidence to the existence of lottery tickets in QNNs. Either discuss and provide evidence in the main part of the paper, but this is a strong claim to add to the conclusion:*** We kindly ask the reviewer to refer to our detailed response to Q6 from Reviewer KsaG.
>
> ***(Q8): Add references to AQC and QUBO introductions; it is a bit difficult for me to understand why the authors made the decision to only give an introduction to QNNs (from a very basic perspective) in the main part of the paper, and not do the same for QUBO and AQC. I'd suggest giving a very brief introduction there as well, or at least, link to the Appendix:*** We have restructured the draft and put the introduction to both QUBO and AQC in section 3 for better readability.

---

> > ### Author Response · Authors · 2025-11-28
> >
> > ***(Q9): Further, I think the introduction to QNNs does not clearly draw a distinction between their setup and the general definition, i.e., QNNs (at least hardware-efficient) often do not use CR operations (your setup), but work with CNOT, CZ. Further, other types of feature encoding (e.g., re-uploading) can be used too. Lastly, the last sentence in Section 3 needs elaboration, it is not clear to me what this should mean:*** we acknowledge the diversity of ansatzes and feature encoding strategies beyond our specific implementation (e.g., different entangling gates like CNOT/CZ, or data re-uploading techniques). However, there has not been standard pre-trained QNNs available which our method can directly use. We therefore, restrict our analysis to one common ansatz which we evaluated empirically. Our goal is to demonstrate the effectiveness of the proposed pruning framework rather than to benchmark all possible QNN designs. The final sentence in Section 3 intends to draw a conceptual analogy to classical networks.
> >
> > ***(Q10): The cost function and expectation value are used interchangeably:*** we have revised the manuscript to use these terms distinctly.
> >
> > ***(Q11): Why is the removal and parameter nullification protocol a result of the Lie group framework:*** the connection stems from the mathematical structure of parametrized quantum gates. The parameterized quantum gates used in QNNs form a Lie group structure. In this framework, each gate $U_k(θ_k) = e^{-iθ_k G_k}$ is generated by $G_k$ from the corresponding Lie algebra. The identity operator $I$ serves as the group identity. Setting $θ_k → 0$ (parameter-nullification protocol) directly yields $U_k(θ_k) → I$ due to the continuity of the exponential map in the Lie group structure. This mathematical foundation rigorously justifies equating gate removal with parameter nullification. We have clarified this connection more explicitly in the revised manuscript in Appendix B.
> >
> > ***(Q12): What does remark 4 mean:*** we give remark 4 to emphasize that our QuBS protocol is not only valid for pure QNNs, but also for hybrid frameworks that uses classical neural networks within a hybrid pipeline.
> >
> > ***(Q13): Where does the ½ factor in Eq.46 go?*** The ½  factor is absorbed into the definition of matrix Q during the algebraic transformation to the standard QUBO form.
> >
> > ***(Q14): What is the difference between Appendix P and Figure 3?*** Figure 3 provides a schematic overview of the hybrid quantum-classical model learning process. For each epoch, we provide the metric value of the whole hybrid model and the model with the QNN masked out; data encoding strategy is also included. Appendix P, instead, presents a study analyzing the training dynamics and performance of the classical backbone feature extractor alone, comparing it against the full hybrid model.
> >
> > ***(Q15): Figure 8: what does this figure depict? What is the x-axis?*** Figure 8 depicts the performance comparison between QuBS and the numerical ground truth obtained via exhaustive search for a small benchmark QNN, where the x-axis represents the combinatorial configuration index; exhaustive search is only possible for small scale QNNs due to exponentially increasing overheads.
> >
> > ***Why is the discrepancy between original and pruned network so big, even for hardly any components removed?*** We think there might be a misunderstanding. In our experiment section, Figure. 4 and 5 consistently show that when a large part of the components are removed from the circuit, the performance drops significantly. For example, in Figure 4 (left), when 90 percent of the gates are removed, the accuracy drops from 80 to 65 percent. As more components are removed, we show that QuBS can find a better sparse QNN compared to existing methods; see Figure 4 and 5.

---

### Official Review · Reviewer_ZbsX · 2025-10-28

**Soundness:** 1
**Presentation:** 2
**Contribution:** 1
**Rating:** 2
**Confidence:** 5

**Summary:**

The authors propose a pruning algorithm for quantum neural networks (QNNs) that leverages the Hessian matrix of the QNN. They formulate the problem of selecting a pruning mask as a QUBO problem, which is then solved using quantum annealing. The authors aim to achieve a computational advantage in solving the QUBO problem.

**Strengths:**

The figure and the presentation look good.

**Weaknesses:**

Weakness:
- There is a fundamental issue with the results of Proposition 1 (Eqn. (7)), which serves as the theoretical foundation of this work.  Specifically, in Eqn. (40), which is a crucial step in deriving Eqn. (7), the third equality only holds if all the generators $G_i$ and $G_j$ commute. However, the authors consider a general quantum circuit without making this strong assumption. As a result, the conclusions in Eqn. (7) are incorrect.
- Some notations in Equation (9) are not well defined. For example, the authors should provide explanations for the terms $\delta w_{pr}$
 and $w_0$.
- There is an inconsistency in the notation used throughout the manuscript. For instance, the parameter is denoted as $\theta$ in Section 4, but as $w$ in Section 5. This lack of consistency should be addressed for clarity.

**Questions:**

The questions are included in the Weakness.

---

> ### Author Response · Authors · 2025-11-28
>
> ***(Q1): There is a fundamental issue with the results of Proposition 1, which serves as the theoretical foundation of this work. Specifically, in Eqn (40), which is a crucial step in deriving Eqn. (7), the third equality only holds if all the generators commute. However, the authors consider a general quantum circuit without making this strong assumption. As a result, the conclusions in Eqn. (7) are incorrect:*** We acknowledge the mistake and have corrected it in the corresponding sections. Importantly, this correction does not affect our proposed approach: the analysis in question was used only as preliminary context and is not relied upon by QuBS. In general, QuBS is designed for pruning QNNs that naturally exhibit dense correlations when quantum gates are removed, and its methodology does not depend on the specific analytical condition previously stated. The corrected analysis now more clearly explains the general circumstances under which such dense correlations vanish, in which case the pruning search complexity would grow only linearly rather than exponentially.
> Should the reviewer have further questions or wish to discuss the technical details, we would be happy to engage in a more in-depth exchange.
>
> ***(Q2): Some notations in Equation (9) are not well defined. For example, the authors should provide explanations for the terms and. There is an inconsistency in the notation used throughout the manuscript. For instance, the parameter is denoted as in Section 4, but as in Section 5. This lack of consistency should be addressed for clarity:*** We have revised the complete manuscript and resolved these notational inconsistencies as suggested.

---

### Official Review · Reviewer_y9p5 · 2025-10-31

**Soundness:** 2
**Presentation:** 2
**Contribution:** 3
**Rating:** 2
**Confidence:** 4

**Summary:**

This manuscript presents a quantum circuit pruning method named QuBS. It formulates the search for an optimal sub-circuit that maintains high fidelity as a Quadratic Unconstrained Binary Optimization (QUBO) problem, solvable via quantum annealers. Experimental results demonstrate the method’s performance under different sparsity ratios and its advantages over two existing algorithms.

**Strengths:**

- The manuscript clearly describes the proposed method, providing both theoretical derivations and algorithmic details.

- The method includes two versions—gate-level pruning and depth-level pruning—as well as an iterative variant for large-scale circuits, making the work more comprehensive.

**Weaknesses:**

- The overall organization of the manuscript is weak; too many sections are created, and the logical connections between them are unclear.

- The writing quality needs improvement. For example, the exact definition of the objective function $\mathcal{L}$ is missing.

- The motivation is questionable. The paper claims to address the misalignment between Euclidean-space heuristics and Hilbert-space dynamics in quantum circuits, but this statement is confusing. A more detailed discussion of these differences and the limitations of existing methods is needed.

- The experimental comparison is insufficient. Only two existing methods are considered, while other pruning approaches, such as symmetric pruning, should also be included. In addition, the circuit scales used in the experiments are too small to convincingly demonstrate scalability.

- Experimental details are missing and should be clearly reported.

**Questions:**

What is the computational complexity of estimating the Hessian matrix?

---

> ### Author Response · Authors · 2025-11-28
>
> ***(Q1): The overall organization of the manuscript is weak; too many sections are created, and the logical connections between them are unclear:*** We have revised the manuscript and give details on how each section is structured; see start of Section 5.
>
> ***(Q2): The writing quality needs improvement. For example, the exact definition of the objective function is missing:*** The definition of our objective is explained in lines 367-370 of the updated draft.
>
> ***(Q3): The motivation is questionable. The paper claims to address the misalignment between Euclidean-space heuristics and Hilbert space dynamics, but this statement is confusing. A more detailed discussion of these differences and the limitations of existing methods is needed: *** We clarify here as follows: existing pruning heuristics (summarized in Table 1) operate in Euclidean parameter space, relying on the assumption that the contribution of each parameter is independent and linear.
> However, in the Hilbert space where quantum states evolve, the effect of removing a gate is governed by the non-commutative nature of quantum operations and it is highly non-linear. This introduces complex, correlated pruning decisions that are inherently combinatorial. By formulating the pruning task as a QUBO problem, QuBS directly optimizes over these correlated decisions, thereby aligning the pruning process with the true Hilbert-space dynamics of the QNN. We have expanded this discussion in the introduction section to clearly articulate these fundamental differences.
>
> ***(Q4): The experimental comparison is insufficient. Only two existing methods are considered, while other pruning approaches, such as symmetric pruning, should also be included:*** Our work specifically targets the general-purpose QNN pruning problem, which imposes no constraints on circuit architecture. The mentioned symmetric pruning approach “Symmetric Pruning in Quantum Neural Networks” [Wang et al, 2022] is not applicable here as they require specific, symmetric circuit ansatzes while we focus on the general QNN architectures. The two baselines (NR-QNN and QAdaPrune) we selected are, to our knowledge, the only existing methods designed for this general setting.
>
> ***The circuit scale used in the experiments are too small to convincing demonstrate scalability:***  Concerning the circuit scale, while the 5-qubit circuit may appear modest, it contains 50 parametrized gates under the preset circuit depth and the ansatz pattern, creating a combinatorial search space of 2^{50) possible pruning configurations—a size already intractable for classical solvers such as exhaustive search. Furthermore, as our study requires pre-training the QNN on a classical simulator (an exponentially expensive process in the absence of standard pre-trained QNNs), the circuit scale is pragmatically constrained. For these reasons, we believe the chosen setup already provides a meaningful and non-trivial benchmark for evaluating our method's efficacy. In the meantime, we include additional experiments involving 10 qubits with reduced circuit depth and include the results in the revision; see Appendix Q.
>
> ***(Q5): Experimental details are missing and should be clearly reported:*** Detailed information regarding the learning setup, ansatz structure, and evaluated applications are provided in Section 6 (QNN architecture and training protocol) and Appendix K. We have checked and made sure all the references are included.

---

### Official Review · Reviewer_KSaG · 2025-11-01

**Soundness:** 3
**Presentation:** 3
**Contribution:** 2
**Rating:** 2
**Confidence:** 4

**Summary:**

This manuscript provides a pruning framework for variational quantum circuits that models  non-commutativity among gates. It uses second order Taylor expansion which gives a QUBO with gradients and Hessian. The authors solve the QUBO on a D-Wave annealer. The theory is complete, and the experiments on a small hybrid model for CIFAR-10, QuBS give more accurate results then other approaches.

**Strengths:**

1. The manuscript provide a good theory framework
2. Hessian has a clean commutator–anticommutator form.
3. The experiments shows consistent gains at high sparsity.

**Weaknesses:**

1. Comparisons are limited to NR-QNN and QAdaPrune. Maybe it should include all methods receive equivalent post-pruning fine-tuning or hyperparameter sweeps.
2. The paper does not report wall-clock costs or budget vs. accuracy trade-offs for this step on larger $d$; exact forward-only Hessians require $O(d^2)$ circuit evaluations.
3. The paper does not compare against strong classical QUBO solvers beyond simulated annealing.
4. The paper works with a small 5 qubit anstz. It is not clear how it scales to larger circuits.

**Questions:**

1. The Lagrangian penalty enforces sparsity but also adds a rank-1 perturbation to the coupling matrix. would there be a strong penalty?
2. Can authors make comments more on "lottery tickets in QNN settings" in the summary?

---

> ### Author Response · Authors · 2025-11-28
>
> ***(Q1): Comparisons are limited to NR-QNN and QAdaPrune:*** In the context of general QNN pruning, to the best of our knowledge, NR-QNN and QAdaPrune are the only two existing methods that address the same general pruning scenario as our work. The work “Quantum Neural Network Compression” [Hu et al, 2022], also includes pruning set-ups as we cited in Sec. 2; however, it adopts the same strategy as in NR-QNN, i.e. the Pauli gate rotation strength is used as the pruning indicator.
>
> ***Inclusion of methods that receive equivalent post-pruning fine-tuning or hyperparameter sweeps:*** Our method targets pruning QNNs to reduce coherence errors when executed on quantum hardware; it does not include the post-pruning fine-tuning phase. Regarding hyperparameter sweeps, our method only introduces a single hyperparameter that is used for enforcing sparsity—the Lagrangian penalty coefficient (see Sec. 5.3). Following established practices in literature (see cited work in lines 307-310), we set the penalty coefficient to the smallest value sufficient to enforce the sparsity constraint, thereby avoiding over-regularization. For all baseline methods, hyperparameters are kept consistent with those reported in their original publications to ensure their best-performing configurations.
>
> ***(Q2): Missing report of wall-clock costs or budget vs. accuracy trade-offs on larger dimensions d:*** The wall-clock time includes two parts: 1) classical problem embedding and 2) annealing time. The wall-clock time for embedding the classical problem of different d (QUBO sizes) into quantum hardware is available in Figure 6-(c). Annealing time on the QPU is calculable from the parameters provided at the start of Section 6 (100 reads × 20 μs per read). When using the iterative variant of QuBS, the accuracy of solving the same problem with different hardware resource requirements has also been recorded in Figure 6-(a, b).
>
> ***Exact forward-only Hessian requires O(d^2) circuit evaluations:*** This observation is correct; this is required for constructing the QUBO matrix of d^2 entries, which implicitly encodes the landscape of 2^d combinatorial configurations.
>
> ***(Q3): The paper does not compare against strong classical QUBO solvers beyond simulated annealing:*** The primary objective of our study is to introduce a forward-looking framework that can leverage potentials (return a high-quality solution faster) of quantum solvers for QNN pruning. Quantum annealing hardware has made considerable progress over the last years in coherence time, solution quality, connectivity, and noise; see work “Quantum Speedup by Quantum Annealing” [Somma et al., 2012]. We expect this progress to continue, which will provide a highly competitive solver for QUBO problems in the future. This work aims to gain such insights into understanding the right problem formulations already today.
>
> Our comparison with simulated annealing (SA) serves to provide a standardized benchmark and does not aim to show that current annealing hardware (still in early stages of development) performs better than classical solvers in terms of absolute solution quality: we show that in Table 2 that the quality of the solution returned by the annealer does not yet surpass strong classical solvers such as simulated annealing even though classical solvers have worse computational complexity scaling for such problems. To complement our work, we have included additional results with classical solvers such as Gurobi in Section. 6.3.
>
> ***(Q4): The paper works with a small 5-qubit ansatz:*** The computational complexity of the pruning problem is not governed by the number of qubits alone, but also the ansatz structure and its depth; see Figure 7. These factors together determine the number of parametrized gates, and therefore the computational complexity and defines the size of the combinatorial search space. Our setup, with a 5-qubit ansatz of 50 parameterized operations, presents a substantial optimization challenge over a search space of 2^50 possible configurations. Therefore, we believe that benchmarking our QuBS framework on a problem of this scale already represents a meaningful and non-trivial validation of its capabilities. As a complement, we included additional results with a circuit of 10 qubits in Appendix R.

---

> > ### Author Response · Authors · 2025-11-28
> >
> > ***It is not clear how it scales to larger circuits:*** The QuBS framework itself is, by design, fundamentally agnostic to problem scale and can be formulated for circuits of arbitrary size. Possible limitation in the scaling is not a problem of our proposed method but the qubit capacity of today’s early-stage quantum annealers; it constrains the size of the QUBO problem that can be solved in a single step. With more mature hardware, a larger scale problem can be accommodated. We also propose an alternative solution to this hardware-imposed constraint by proactively introducing an iterative local pruning variant (Section 5.5). This variant decomposes the global pruning problem into manageable sub-problems, enabling applications with lower requirement of hardware resources.
> >
> > ***(Q5): The Lagrangian penalty enforces sparsity but also adds a rank-1 perturbation to the coupling matrix. Would there be a strong penalty?:*** It is correct that this method introduces a rank-1 perturbation. However, this is a well-characterized technique in optimization literature; see cited works in L288-290, and its effects are manageable. Crucially, our implementation adheres to the established principle of setting the penalty coefficient to the minimum value required to enforce the sparsity constraint, a strategy explicitly designed to avoid over-regularization and minimize any undue distortion of the original QUBO problem. The empirical evidence strongly supports this approach: as demonstrated in Section 6, QuBS consistently outperforms prior methods in preserving accuracy at high sparsity levels. This confirms that the perturbation is effectively controlled and does not compromise the practical performance of our framework.
> >
> > ***(Q6): Can authors make comments more on “lottery tickets in QNN settings” in the summary:*** Our statement that QuBS "provides evidence for the existence of lottery tickets in QNN settings" is a direct empirical extension of the Lottery Ticket Hypothesis (LTH)  into QNNs. The core claim of the LTH is that dense networks contain sparse, trainable subnetworks capable of matching the original performance, which is the core rationale behind pruning. Through QuBS, we have empirically identified precisely such subnetworks within dense QNNs, as evidenced by the highly-sparse circuits in Figures 4 and 5 that maintain high accuracy without any post-pruning fine-tuning. The successful isolation of these performant sparse structures directly parallels the central finding of the classical LTH, thereby providing compelling evidence for the existence of "lottery tickets" in QNNs.

---

### Author Response · Authors · 2025-11-28

We greatly thank the reviewers for providing valuable feedbacks and letting us know in what parts we can still improve and what parts we have not structured well. We have carefully revised the manuscript to address the raised concerns and updated the corresponding draft; all changes are highlighted in red.

---

### Note · Program_Chairs · 2026-01-17
**Submission Desk Rejected by Program Chairs**

The following references in this submission do not refer to real documents and/or have major errors in bibliographic information:

 Evan Jeffrey Zhang, Hsin-Yuan Huang, Richard Chen, Yingkai Ma, Richard Kueng, and John Preskill. Benchmarking noisy intermediate-scale quantum devices. PRX Quantum, 4(1):010328, 2023. doi: 10.1103/PRXQuantum.4.010328.